# Geostatistical interpolation by Quantile Kriging

Henning Lebrenz[1,2] and András Bárdossy[2]

[1]University of Applied Sciences and Arts - Northwestern Switzerland, Institute of Civil Engineering, Switzerland
[2]University of Stuttgart, Institute for Modelling Hydraulic and Environmental Systems, Germany

**Correspondence:** Henning Lebrenz (henning.lebrenz@fhnw.ch)

**Abstract.** The widely applied geostatistical interpolation methods of Ordinary Kriging (OK) or External Drift Kriging (EDK) interpolate the variable of interest to the unknown location, providing a linear estimator and an estimation variance as measure of uncertainty. The methods implicitly pose the assumption of Gaussianity on the observations, which is not given for many variables. The resulting 'best linear and unbiased estimator' from the subsequent interpolation optimizes the mean error over many realizations for the entire spatial domain and, therefore, allows a systematic under- (over-) estimation of the variable in regions of relatively high (low) observations. In case of a variable with observed time-series, the spatial marginal distributions are estimated separately for one time step after the other, and the errors from the interpolations might accumulate over time in regions of relatively extreme observations.

Therefore, we propose the interpolation method of Quantile Kriging (QK) with a two step procedure prior to interpolation: we firstly estimate distributions of the variable over time at the observation locations and then estimate the marginal distributions over space for every given time step. For this purpose, a distribution function is selected and fitted to the observed time-series at every observation location, thus converting the variable into quantiles and defining parameters. At a given time step, the quantiles from all observation locations are then transformed into a Gaussian-distributed variable by a twofold quantile-quantile transformation with the Beta- and the Normal-distribution function. The spatio-temporal description of the proposed method accommodates skewed marginal distributions and resolves the spatial non-stationarity of the original variable. The Gaussian-distributed variable and the distribution parameters are now interpolated by OK and EDK. At the unknown location, the resulting outcomes are reconverted back into the estimator and the estimation variance of the original variable. As a summary, QK newly incorporates information from the temporal axis for its spatial marginal distribution and subsequent interpolation and, therefore, could be interpreted as a space-time version of Probability Kriging.

In this study, QK is applied for the variable of observed monthly precipitation from raingauges in South Africa. The estimators and estimation variances from the interpolation are compared to the respective outcomes from OK and EDK. The cross-validations shows that QK improves the estimator and the estimation variance for most of the selected objective functions. QK further enables the reduction of the temporal bias at locations of extreme observations. The performance of QK, however, declines when many zero-value observations are present in the input data. It is further revealed that QK relates the magnitude of its estimator with the magnitude of the respective estimation variance as opposed to the traditional methods of OK and EDK, whose estimation variances do only depend on the spatial configuration of the observation locations and the model settings.

# 1 Introduction

Many environmental variables (e.g. precipitation, ore grades) are only measured at some distinct observation locations, but possess a highly variable and unknown spatial distribution (Armstrong, 1998). The regionalization of the variable, i.e. the interpolation of the variable to the unknown locations, attempts the full description of its spatial distribution as a prerequisite for practical objectives (e.g. hydrological modeling, efficient exploitation of resources). However, a deterministic description of the spatial distribution is severely hampered for many variables since they incorporate a complex genesis which is neither fully known nor understood.

Therefore, the assessment of the distribution by geostatistical models arose, whose theoretical fundamentals were firstly laid out by Matheron (1962). The theory regards the (observed) values of a variable $z$ as one realization $z(\mathbf{x})$ of a Random Variable (RV) $Z$ at the specific location $\mathbf{x}$ ($= \{x_1, x_2\}$ for $\mathbb{R}^2$). Since the variable is often only partially known at a few distinct measurement locations $\mathbf{x}_i$, ergodicity is assumed and the intrinsic hypothesis for Ordinary Kriging (OK, Matheron (1965)) is given as:

$$E[Z(\mathbf{x})] = m(\mathbf{x}) = \text{const.} \tag{1}$$

$$VAR[Z(\mathbf{x}+\mathbf{h}) - Z(\mathbf{x})] = 2\gamma(\mathbf{h}) \tag{2}$$

in which $\gamma(\mathbf{h})$ is the semi-variogram. The increment $Z(\mathbf{x}+\mathbf{h}) - Z(\mathbf{x})$ is assumed as a stationary random function and its variance only depends on the translation vector $\mathbf{h}$. The set of outcomes by the interpolation to the unknown location $\mathbf{x}$ is described by the linear estimator $Z^*(\mathbf{x})$ as a measure of their centrality and by the estimation (or Kriging) variance $\sigma_K^2(\mathbf{x})$ as a measure of the associated uncertainty.

The stated hypothesis entails three implications: the first condition of the intrinsic hypothesis (Eq.(1)) demands the variable to be spatially stationary and yields an unbiased estimation error in the entire domain (Chilès and Delfiner, 1999). Therefore, a systematic under- (over-) estimation is induced for regions of high (low) observations. Secondly, the marginal distribution of the observed data should ideally be Gaussian in order to be adequately described. Unfortunately, the distribution often departs from the ideal case (Journal and Alabert, 1989), necessitating an a-priori transformation of the marginal distribution. And last, the second condition (Eq.(2)) implies that the magnitude of the Kriging variance $\sigma_K^2(\mathbf{x})$ only depends on the spatial configuration of the observation locations, the a-priori variance of all observations and on the selected variogram model, but not on the magnitude of the linear estimator $Z^*(\mathbf{x})$ itself (Goovaerts, 2000).

The theoretical extension of External Drift Kriging (EDK, Ahmed and deMarsily (1987)) addresses the first implication. EDK can be attributed to the non-stationary geostatistical interpolation methods and has been frequently applied in various disciplines of practice and science (e.g.: Bourennane et al. (2000), van de Kassteele et al. (2009), Motaghian and Mohammadi (2011)). It incorporates additional information from external variables (or drifts) $Y_j(\mathbf{x})$ for the estimation of the variable at the

unknown location. The mean $m(\mathbf{x})$ is non-stationary but linearly depends on the external variables. Thus, the first condition (Eq.(1)) is reformulated to:

$$E\left[Z(\mathbf{x})\right] = a + \sum_{i=1}^{k} b_i \cdot Y_i((\mathbf{x})) \qquad (3)$$

where $k$ is the number of the incorporated external drifts $Y_i(\mathbf{x})$, while $a$, $b_i$ are the unknown constants. The drifts are required to be known prior to interpolation at all relevant locations. Ideally, EDK still requires a marginal Gaussian distribution.

The non-parametric methods of Indicator Kriging and Probability Kriging ( Journel, 1983) are further derivatives and allow an a-priori transformation of the skewed marginal distribution. Indicator Kriging transforms the variable $z$ into a binary variable by defining a threshold value $z_{th}$ and restates the first condition (Eq.(1)) of the random function $I(\mathbf{x}; z_{th}) = 1_{Z(\mathbf{x}) < z_{th}}$ to:

$$E\left[I(\mathbf{x}; z_{th})\right] = m = const. \qquad (4)$$

This non-linear transformation is relatively robust and limits the effect of high values on the description of the variable at the unknown location. However, a loss of information comes along by the transformation into a binary variable. Therefore, Probability Kriging defines multiple thresholds and uses the order relation of the observed variable ( Carr and Mao, 1993), being implemented by using co-kriging in the derived multi-variate context. Both non-parametric methods have been subject to research, especially for detection limit problems of groundwater contamination (e.g. Goovaerts et al. (2005), Lee et al. (2007), Adhikary et al. (2011)).

In summary, geostatistical methods have been derived in the past in order to address the stated shortfalls of the intrinsic hypothesis. However, all present methods only regard the observations from the one respective time step for the estimation of their marginal spatial distribution, but do not incorporate observations from other time steps. The inclusion of a temporal behavior into the geostatistic models is mostly irrelevant for the original geological variables. However, the temporal variability of a variable becomes more prominent for other sciences, e.g. hydrology, where observations from raingauges over several time steps are implemented into the geostatistical models in order to generate spatial precipitation estimates. These estimates subsequently serve as input to the hydrological modelling (e.g. Syed et al. (2003), Basistha et al. (2008), Cole and Moore (2008)) over multiple time steps. Associated errors in the precipitation estimates may ultimately lead to greater errors in the subsequent discharge modelling (Kobold and Sušelj, 2005). These errors strongly depend on the spatial and temporal distribution of the input precipitation (Gabellani et al. (2007), Moulin et al. (2009)) and may limit the accuracy of rainfall-runoff simulations. There are geostatistical space-time models in order to incorporate the temporal variability of the variable, but they are primarily aiming on the extrapolation of the variable in time ( Snepvangers et al. (2003)). Therefore, they require a strong dependence of the variable over time, suited e.g. for groundwater modeling where temporal changes occur relative slowly. This temporal dependence might be absent for other variables, e.g. monthly precipitation.

In the following section, we introduce Quantile Kriging as a spatio-temporal description of the variable $Z$, addressing the three shortfalls: non-stationary variables, skewed marginal distributions over space and the independence of the error distribution from the magnitude of the observation.

## 2  Materials and Methods

The theory of Quantile Kriging (QK) is outlined along with the major theoretical implications, followed by a general discussion of the underlying geostatistical model and a case study for the variable of monthly precipitation is presented.

FIGURE 1

A preliminary analysis of the selected variable exemplary reveals (Fig. 1) the non-Gaussianity within the data and that the first assumption of the intrinsic hypothesis (see Eq.(1)) is not fulfilled since e.g. $E_t[Z(\mathbf{x}_{57}, t)] \neq E_t[Z(\mathbf{x}_{29}, t)]$.

### 2.1  Theory of Quantile Kriging

QK presumes the existence of observations of the variable $z$ over consecutive time steps t $(= 1, 2, ..., J)$ at every observation location $\mathbf{x}_i = (x1_i, x2_i)$ [for the two-dimensional space $\mathbb{R}^2$], providing an observed time series $z(\mathbf{x}_i, t)$ at every observation location i $(= 1, 2, ..., n_i)$. QK proceeds first with a two step procedure prior to interpolation (see Sect. 2.1.1) and second with the interpolation itself (see Sect. 2.1.2):

### 2.1.1  Estimation of the temporal and the spatial marginal distribution

At first, the distribution over time is estimated at every observation location location $\mathbf{x}_i$: an appropriate theoretical cumulative distribution function (cdf) $F$ is selected and fitted to the corresponding time series of observations $z(\mathbf{x}_i, t)$, yielding $n_i$ specific distributions $F(z(\mathbf{x}_i, t))$. The distributions are defined by their corresponding parameter sets $\Theta(\mathbf{x}_i)$ $(= \vartheta_k(\mathbf{x}_i)$ with $k = 1, 2, ..., K)$ of the $K$-parametric distribution function $F$. The quantiles $w(\mathbf{x}_i, t)$ $(= F(z(\mathbf{x}_i, t); \Theta(\mathbf{x}_i)))$ are calculated from the observed values of the variable $z(\mathbf{x}_i, t)$ and the defining parameter set $\Theta(\mathbf{x}_i)$. The quantiles $w(\mathbf{x}_i, t)$ possess a uniform distribution over time on the interval $[0, 1]$ for a given observation location $\mathbf{x}_i$. However, their empirical distribution in space is not uniform on $[0, 1]$. In order to profit from the optimality of Kriging, it requires a transformation into a Gaussian distribution as a prerequisite of the subsequent interpolation.

The marginal spatial distribution, corresponding to a time step t is, therefore, estimated by a twofold quantile-quantile conversion as the second step: the two-parametric Beta- distribution is fitted to the quantiles $w(\mathbf{x}_i, t)$ of a given time step t, whose cdf $G(w; \alpha, \beta)$ is defined as:

$$G(w; \alpha, \beta) = \sum_{n=\alpha}^{\alpha+\beta-1} \left[ \frac{(\alpha+\beta-1)!}{n! \cdot (\alpha+\beta-1-n)!} \cdot w^n \cdot (1-w)^{(\alpha+\beta-1-n)} \right] \tag{5}$$

on the interval $[0,1]$ by the two parameters $\alpha > 0$ and $\beta > 0$. The quantiles $G(w(\mathbf{x}_i,t);\alpha(t),\beta(t))$ from Eq. (5) are finally transformed by a Normal Score Transformation into the standard Gaussian variable $u(\mathbf{x}_i,t)$ with $N_u[0|1]$, which ultimately serves as spatial marginal distribution to the subsequent geostatistical interpolation.

The transformation via the quantiles into the variable $u$ accounts for spatially non-stationary distributions of the original variable $z$ with $E[Z(\mathbf{x},t)] \neq m$ and exchanges the two conditions of Eq. (1) and (2) to:

$$E[F_x(Z(\mathbf{x},t))] = m = \mathrm{const.} \tag{6}$$

$$VAR[F_{x+h}(Z(\mathbf{x}+\mathbf{h},t)) - F_x(Z(\mathbf{x},t))] = 2\gamma(\mathbf{h}) \tag{7}$$

resolving the problem of spatial non-stationarity. QK can accommodate skewed marginal distributions of the original variable $Z$, which is similar to Probability Kriging, but it newly incorporates the temporal behavior of $Z$ into its estimation of the spatial marginal distribution.

### 2.1.2 Interpolation to the unknown location

The outlined conversion of the variable $z(\mathbf{x}_i,t)$ into the variable $u(\mathbf{x}_i,t)$ and its corresponding parameter set $\Theta(\mathbf{x}_i)$ entails separate interpolations to the unknown location $\mathbf{x}$.

The inherent assumption of second-order stationarity implies the existence of a constant spatial mean for the variable $u$ within the domain for every time step $t$. The transformed quantiles are implicitly assumed to be more homogeneously distributed over space than the original variable $z$. The variable $u$ is subject to a stationary geostatistical interpolation method (e.g. OK), providing a linear estimator $U^*(\mathbf{x},t)$ and the estimation variance $\sigma^2_{K,U}(\mathbf{x},t)$. They jointly describe the Gaussian distribution of the random variable $U(\mathbf{x},t)$ with $N[U^*|\sigma^2_{K,U}]$.

The defining parameters $\vartheta_k$ (for $k = 1,2,...,K$) of the K-parametric distribution function $F$ are independent from the time step $t$ and they are separately interpolated to the unknown location $\mathbf{x}$. The separate interpolation, however, requires the independence of the parameters $\vartheta_k$ from each other. Therefore, a Principal Component Analysis examines the 'observed' parameters $\vartheta_k(\mathbf{x}_i)$ in the Cartesian coordinate system $(\vartheta_1|\vartheta_2|...|\vartheta_K)$ and determines the corresponding rotation angle $\alpha$ and translation vector $\mathbf{k}$. The coordinate system is then rotated and translated accordingly prior to interpolation in order to ensure independence. A possible spatial non-stationarity of the parameters can be accounted for by the choice of an appropriate non-stationary interpolation method (e.g. EDK). The interpolation of the independent parameters yields their estimators at the unknown location $\mathbf{x}$, which are rotated and translated back to the original coordinate system. Thus, the estimators $\vartheta^*_k(\mathbf{x})$ are defining the distribution function at the unknown location $\mathbf{x}$.

At last, the resulting Gaussian distribution of the random variable $U(\mathbf{x},t)$ is reconverted into a distribution of the original variable $Z(\mathbf{x},t)$ by the outlined steps of conversion (Section 2.1.1), but in reverse order: first the distribution of the quantiles $G(W(\mathbf{x},t);\alpha,\beta)$ of the Beta-distribution are calculated by the inverse of the Normal Score Transformation. The distribution of the quantiles $W(\mathbf{x},t)$ are calculated next using the inverse of Eq. (5) and last, the distribution of the original variable $Z(\mathbf{x},t)$ is estimated by using the inverse of the selected cdf, being defined by the estimators of its parameters $\vartheta^*_k(\mathbf{x})$. The reconversion

of the distribution of $U(\mathbf{x}, t)$ to the distribution of $Z(\mathbf{x}, t)$ can be implemented by the simple numerical Rosenblueth point estimation method (Rosenblueth, 1975). The resulting distribution of the original variable $Z$ is then described by the expectation value $Z^*(\mathbf{x}, t)$ and the variance $\sigma_K^2(\mathbf{x}, t)$. Note that the resulting asymmetrical distribution of $Z(\mathbf{x}, t)$ is non-Gaussian due to the conversion with the non-linear, but monotonic theoretical cdf $F$.

The basic methodology of the proposed QK is illustrated in Fig. 2.

FIGURE 2

## 2.2  Discussion of the geostatistical model

Since the proposed method of QK is applied for the variable of monthly precipitation (see Chapter 2.3), the discussion of the underlying process model is based on the following properties of precipitation fields:

– The monthly (and even daily) precipitation amounts $z(\mathbf{x}, t)$ for a given time step $t$ often show a skewed distribution and cannot be considered as stationary over space. The differences in expected precipitation amounts become especially obvious for long time accumulations.

   – The meteorological processes, which are generating precipitation, are usually of large spatial extent: if one location receives heavy precipitation, it is likely that other locations also receive heavy precipitation.

– Correlations between time series of precipitation indicate a strong spatial dependence, while the spatial dependence of precipitation at one given time step (e.g. day, month) usually show a much weaker spatial dependence.

   A possible process model reflecting the above properties can be described as follows:

   Let $U(\mathbf{x}, t)$ be independent (for each different time step $t$) normal stationary spatial fields with $E[U] = 0$ and $D^2(U) = 1$ for each time step $t$. Now, the process $M(t)$ is introduced in order to reflect large scale meteorological processes. High $M(t)$

values correspond to heavy precipitation covering the area, while low values correspond to dry conditions, as it is reflected by seasonal variations of precipitation amounts. The introduced $M$ modifies the spatial process to:

$$G(\mathbf{x}, t) = U(\mathbf{x}, t) + M(t) \tag{8}$$

   where $M(t)$ is a process (only in time) with a mean of zero. We may assume that the distribution of $M(t)$ is normal and, therefore, $G(\mathbf{x}, t)$ would be normally distributed with $N(0, d)$ (with $d = \sqrt{1 + \sigma_M^2}$) at every location $\mathbf{x}$.

For each individual time step $t$, the distribution of $G(\mathbf{x}, t)$ is $N(M(t), 1)$ and the resulting spatial field $W$ is the temporal non-exceedance probability at location $\mathbf{x}$ being confined to $0 \leq W(\mathbf{x}, t) \leq 1$ and formally described as:

$$W(\mathbf{x}, t) = \Phi_{0,d}(G(\mathbf{x}, t)) \tag{9}$$

where $\Phi_{0,d}$ is the distribution function of $N(0,d)$. The precipitation is then generated as:

$$Z(\mathbf{x},t) = F_x^{-1}(W(\mathbf{x},t)) \tag{10}$$

where $F_x$ is the distribution function of precipitation at the location $\mathbf{x}$. The distribution functions $F_x$ may vary between different locations $\mathbf{x}$ due to topography and other influencing factors, and they could be subject to interpolation (e.g. Mosthaf and Bárdossy

(2017)).

We use $W(\mathbf{x},t)$ for each time step $t$ and assume that it follows a Beta-distribution. In fact, its distribution depends on $M(t)$. If $M(t) = 0$ for all time steps $t$, then monthly precipitation can be fully characterized by independent realizations over space. In this case, the distribution of $W$ is uniform for each $t$.

However, this is not the case with observed data because wet and dry conditions occur simultaneously over the entire domain.

This is controlled by $M(t)$, which can be taken e.g. as an independent random variable or to follow an ARMA process. If $M(t) \neq 0$ then the distribution of $W(\mathbf{x},t)$ is not uniform for this specific time step $t$. The exact form of the corresponding distribution would be something like:

$$G_t(v) = \Phi_{0,1}\left(\Phi_{M(t),1}^{-1}(v)\right) \tag{11}$$

However, the use of Eq.11 would require the estimation of $M(t)$ for each time step $t$. We decided to use a simple Beta-

distribution instead. The reason for assuming a Beta-distribution is due to their flexibility and their ability to describe distributions well within the interval $[0,1]$.

The introduction of $M(t)$ is reasonable as it explains the difference between the correlation between stations and the spatial correlation calculated using a variogram type approach for a given time step. The later correlations are usually lower (smaller ranges) which are increased by the common large scale weather described by $M(t)$. Note that the introduction of $M(t)$ leads

to a correlation of the precipitation time series even if the individual snapshots of $U(\mathbf{x},t)$ are independent in space.

We estimate and subsequently interpolate $F_x$ within the proposed methodology by the preceding conversion of the variable $Z(\mathbf{x}_i,t)$. In addition, we calculate $W(\mathbf{x}_i,t)$ for the observation locations $\mathbf{x}_i$ and interpolate it to the unknown location $\mathbf{x}$ in order to come back to $Z(\mathbf{x},t)$. In here, spatial variograms are calculated for $W$ for each time step $t$, assuming $W$ to be spatially stationary. Non-Gaussian and non-stationary distributions only occur for the precipitation amounts (i.e. the variable $Z$).

Non-Gaussianity should be considered due to the usually skewed distribution of precipitation amounts and it only applies to the marginal distribution at a given time step $t$. The suggested model should enable a simulation of the precipitation amounts. The spatial dependencies are considered to correspond to a multi-Gaussian copula, being a type of transformation frequently used (e.g. for Lognormal Kriging).

The distributions $F_x$, fitted to the individual locations, are supposed to have a spatial dependence. They are further assumed

to follow the same distribution (e.g. $\Gamma$- or Weibull-distribution) and are subsequently interpolated. Inhere, we assume that the large scale meteorological processes, generating precipitation, are better reflected by the distributions than by a single

monthly (or daily) realization. Therefore, the use of external covariates, e.g. elevation, is deemed more appropriate for their interpolation. The usage of these distributions transforms the process into a stationary one, which is then interpolated using the Beta-distribution of the non-exceedance probabilities.

## 2.3 Application of Quantile Kriging

The proposed method of QK is applied for the variable of monthly precipitation in South Africa and the outcomes are compared to those from OK and EDK.

### 2.3.1 Study Area and Data

The rectangular study area $(3.5° \times 3.5°$, Fig. 3) covers approx. $132000 \, km^2$ and is located within the Republic of South Africa. The second release of the digital elevation model from the Shuttle Radar Topography Mission ( USGS, 2003) serves as elevation

input. The original resolution was upscaled from 3 arcseconds (approx. $92 m$) to 2 arcminutes (approx. $3700 m$) by spatial averaging, resulting in a mean of $1442 m$ and ranging from $669 m$ to $2197 m$ (a.m.s.l.).The upscaled elevation ultimately serves as external drift for EDK of the parameters within QK and for the reference EDK with the original variable.

FIGURE 3

The observations of monthly precipitation were retrieved from raingauges of four different sources: the Department of Water

Affairs (DWA, 2008), the Global Historical Climatology Network ( Vose et al., 1992), the Climate Research Unit (Mitchell and Jones, 2005) and the internal database of the University of KwaZulu-Natal (Lynch, 2004). Accumulation of daily recordings yield monthly values for the $264 (= J)$ consecutive months from January $1986$ to December $2007$. A total of $226 (= n_i)$ raingauges (Fig. 3) provided $32226 (= N)$ monthly precipitation values, which ultimately serve as input data.

The observed average monthly precipitation over the twelve calendar months $c$ is illustrated in Fig. 4 along with the percent-

age of zero-value observations over all observations of the specific calender month $c$ (hereafter referred to as the dry ratio), revealing a seasonal variation. High precipitation is typically encountered in the calendar months from October to March, being characterized by a low dry ratio $< 3 \%$. The study area receives relatively low precipitation amounts during the calender months from April (dry ratio $= 11 \%$) to September (dry ratio $= 25 \%$).

FIGURE 4

### 2.3.2 Adaptation to monthly precipitation

At first, we subdivided the observations of monthly precipitation into the corresponding calendar month $c (= 1, 2, ..., 12)$ prior to the fitting of the selected distribution function due to two reasons: the seasonal variation in monthly precipitation (Fig. 4) and to ensure independence of the individual sample members as a theoretical requirement for the fitting method. We used the maximum likelihood estimation method for fitting the selected distribution function to the respective measurements

values $z(\mathbf{x}_i, t_c)$ of every calender month c and every measurement location $\mathbf{x}_i$, resulting in a total of $2712 (= 12 \times 226)$ fittings. In this context, the two-parametric $\Gamma$- and Weibull- distribution were selected, whose cdf $F(z; \Theta)$ are defined as:

$$\Gamma \text{ - distribution :} \quad F(z;\Theta) = \frac{\gamma(\mu, \lambda \cdot z)}{\Gamma(\mu)} \tag{12}$$

$$\text{Weibull - distribution :} \quad F(z;\Theta) = 1 - exp\left[-\left(\frac{z}{\lambda}\right)^k\right] \tag{13}$$

where $\Gamma(\mu)$ is the gamma function and $\gamma(\mu, \lambda \cdot z)$ is the lower incomplete gamma function. The parameter set $\Theta_c(\mathbf{x}_i)$ is composed for the $\Gamma$- distribution out of $\mu_c(\mathbf{x}_i) (= \vartheta_{1,c}(\mathbf{x}_i))$ and $\lambda_c(\mathbf{x}_i) (= \vartheta_{2,c}(\mathbf{x}_i))$ and for the Weibull- distribution out of $k_c(\mathbf{x}_i) (= \vartheta_{1,c}(\mathbf{x}_i))$ and $\lambda_c(\mathbf{x}_i) (= \vartheta_{2,c}(\mathbf{x}_i))$. All parameters are restrained to values greater than zero and both cdfs are defined for $z(\mathbf{x}_i,t) \geq 0$.

Thus, the original observations of monthly precipitation $z(\mathbf{x}_i,t)$ are converted by Eq. (12) or Eq.(13) into the corresponding quantiles $w(\mathbf{x}_i,t) (= F(z(\mathbf{x}_i,t);\Theta_c))$ and their defining parameter set $\Theta_c(\mathbf{x}_i)$. As outlined in Sect. 2.1, the quantiles $w(\mathbf{x}_i,t)$ were further converted into the standard Gaussian variable $u(\mathbf{x}_i,t)$, ultimately subject to the subsequent OK as our chosen geostatistical interpolation method. Note that the stationary assumption of more homogeneously distributed quantiles in space appear more plausible in the case of monthly precipitation. In total, the variable $u(\mathbf{x}_i,t)$ was interpolated $264$ times by OK to the unknown location $\mathbf{x}$. The corresponding variograms are calculated using Kendall's tau for a robust interpolation (Lebrenz and Bárdossy, 2017).

We further selected EDK as non-stationary interpolation method for the defining parameters $\vartheta_{1,c}$ and $\vartheta_{2,c}$. Elevation data (Sect. 2.3.1) are taken as external drift since the distributions of monthly precipitation are assumed to depend on the altitude of the terrain. However, it was revealed that the direct use of the parameters may lead to negative or zero estimators at locations of extreme external drifts. Therefore, the sample mean $\bar{z}_c(\mathbf{x}_i)$ and the sample variance $\sigma_c^2(\mathbf{x}_i)$ are estimated instead, using the two statistical moments of the selected distribution functions, defined as:

$$\Gamma \text{ - distribution :} \quad E[Z] = \frac{\mu}{\lambda}; \qquad\qquad VAR[Z] = \frac{\mu}{\lambda^2} \tag{14}$$

$$\text{Weibull - distribution :} \quad E[Z] = \lambda \cdot \Gamma(1+1/k); \qquad\qquad VAR[Z] = \lambda^2 \cdot \Gamma(1+2/k) - E[Z]^2 \tag{15}$$

The dependence of the two derived parameters $\bar{z}$ and $\sigma^2$ on each other appears obvious in the case of monthly precipitation: a high mean is likely to be associated with a high variance and v.v.. Their dependence is exemplary illustrated for the calender month 'May' in Fig. 5.

FIGURE 5

The principal component analysis allows for the transformation into the new Cartesian coordinate system with the new coordinates $r_c(\mathbf{x}_i)$ and $s_c(\mathbf{x}_i)$. They are now independent and subject to a separate interpolation by EDK. A total of $24 (= 12 \times 2)$ interpolations by EDK to the unknown location $\mathbf{x}$ is performed for each selected type of distribution.

## 3 Results and Discussion

The proposed interpolation method of QK, using either a $\Gamma$-distribution (QK-$\Gamma$) or a Weibull-distribution (QK-Wei), is implemented and compared to the traditional geostatistical interpolation methods of OK and EDK. The respective performances are evaluated by cross-validation for the resulting estimators $Z^*$ and the associated Kriging variances $\sigma_K^2$. In here, cross-validation
eliminates all values $z(\mathbf{x}_i, t)$ in turns from the input data, and subsequently calculates the estimator $Z^*(\mathbf{x}_i)$ and the associated Kriging variance $\sigma_K^2(\mathbf{x}_i)$ from the remaining data. Only the 32226 data points of the actually recorded values were considered for the cross-validation and the resulting outcomes are compared to the actual observations.

### 3.1 Implementation of Quantile Kriging

The outcomes from the interpolation by OK, EDK and QK-$\Gamma$ are exemplary displayed and examined for a month with low
precipitation and a high dry ratio (August 1993) and a month with high precipitation and a low dry ratio (January 1996). The respective spatial patterns of the estimator $Z^*(\mathbf{x})$ and the associated standard deviation $\sigma_K(\mathbf{x})$ are illustrated in Figs. 6 and 7.

FIGURE 6

FIGURE 7

The estimator $Z^*$ displays similar spatial patterns and value ranges for all the interpolation methods. However, the local
contours of the isohyets are more rugged for QK-$\Gamma$ (Figs. 6(e) and 7(e)) than for OK (Figs. 6(a) and 7(a)), but smoother than for EDK (Figs. 6(c) and 7(c)).

QK utilizes elevation for the interpolation of the two distribution parameters $\vartheta_{1,c}$ and $\vartheta_{2,c}$. The two parameters incorporate information from all time steps $t_c$ of the specific calendar month $c$ and, thus, transfer information over time. They are further combined with the ordinary kriged quantiles $W(\mathbf{x}, t)$, leading to more smooth contours of the isohyets than EDK (compare
Fig. 3). We regard the resulting spatial patterns of QK as more plausible, assuming that the accumulated monthly precipitation is hardly affected by local features in elevation.

The standard deviations $\sigma_K$ of the associated estimation error show notable deviations in spatial pattern for the implemented interpolation methods. The range of error is notably higher for QK (Figs. 6(f) and 7(f)) and its spatial patterns deviates from the typical, bull-eye shaped patterns of OK (Figs. 6(b) and 7(b)) or EDK (Figs. 6(d) and 7(d)).
The estimation error from OK and EDK depends on the spatial configuration of the observation locations $\mathbf{x}_i$, their global variance and the selected variogram model. This typical spatial pattern of the error distribution from the ordinary kriged quantiles $W(\mathbf{x})$ is converted within QK by the monotonic cdf (Eq. (12)). The resulting $\sigma_K(\mathbf{x})$ of the original variable $Z(\mathbf{x})$ is, therefore, increased by relatively flat slopes of the cdfs, which are encountered for relatively high values of $W(\mathbf{x})$. A relationship between the magnitude of the estimator $Z^*$ and the magnitude of the associated standard deviation $\sigma_K$ is suggested
by Figs. 6 and 7.

### 3.1.1 Relationship between estimator and standard deviation

A relationship between the magnitude of the estimator $Z^*$ and the associated standard deviation $\sigma_K$ would be possibly an improvement to geostatistical interpolation and is, therefore, examined next by cross-validation. The Spearman rank correlation coefficient $\rho_S$ is chosen for its description, being defined as:

$$\rho_S = \frac{\sum\limits_{i=1}^{n}(rg(Z^*(\mathbf{x}_i,t)) - \overline{rg}_{Z^*}) \times (rg(\sigma_K(\mathbf{x}_i,t)) - \overline{rg}_{\sigma})}{\sqrt{\sum\limits_{i=1}^{n}(rg(Z^*(\mathbf{x}_i,t)) - \overline{rg}_{Z^*})^2 \times \sum\limits_{i=1}^{n}(rg(\sigma_K(\mathbf{x}_i,t)) - \overline{rg}_{\sigma})^2}} \tag{16}$$

where $rg(Z^*(\mathbf{x}_i,t))$ and $rg(\sigma_K(\mathbf{x}_i,t))$ are the ranks of the estimator $Z^*$ and the associated standard deviation $\sigma_K$ within a set of data, while $\overline{rg}_{Z^*}$ and $\overline{rg}_{\sigma}$ are the respective average ranks. The non-parametric Spearman rank correlation $\rho_S$ describes the monotonic relation between the estimator $Z^*$ and estimation standard deviation $\sigma_K$, ranging from $-1$ (negative) to $+1$ (positive) with $0$ indicating its absence. A set of data consists of all $n$ values of the corresponding calendar month $c$. The evolution of the Spearman rank correlation coefficient $\rho_S$ over all 12 calendar months is displayed in Fig. 8.

### FIGURE 8

The rank correlation varies over the calendar months for all implemented interpolation methods and reach their seasonal maximums in June or July (Fig. 8), being characterized by a high dry ratio and low precipitation.

An improvement in the relationship between the estimator $Z^*$ and the associated standard deviation $\sigma_K$ can be observed for QK-$\Gamma$ and QK-Wei, exhibiting superior rank correlation coefficients for all calender months with the exception of QK-Wei in December (Fig. 8). QK-$\Gamma$ deploys the strongest relation during the wetter months from October to March, while QK-Wei is superior from May to September. The resulting spread of the error distribution is increased by decreasing slopes of the theoretical cdfs (Eq. 12 and 13) and v.v.. The slope is effectively the probability density function (pdf). Both selected theoretical distributions imply a monotonic decrease in their respective pdfs for small parameters, being typically encountered during the dry season, and evoke a higher spread of the error distribution for higher monthly precipitation. Thus, the almost perfect rank correlation $\rho_S(c)$ of QK during the months of low precipitation can be explained. The rank correlation between the estimator and the standard deviation is weakened for the months with higher precipitation due to the departure from the strict monotonic decrease of the pdfs, which is induced by increasing distribution parameters.

The inferior correlation coefficients of OK and EDK are nearly congruent due to their inherent geostatistical definition: although the Kriging weights are altered by the drift, they influence the linear estimator $Z^*$ and the standard deviation $\sigma_K$ by the same extent. Therefore, the non-parametric Spearman descriptor hardly differentiates between OK and EDK.

### 3.2 Cross-validation of the estimator

The estimator $Z^*(\mathbf{x}_i,t_j)$ from the cross-validation is evaluated by six objective functions: the Pearson correlation coefficient $\rho$, the Nash-Sutcliffe-Efficiency coefficient NSE (Nash and Sutcliffe, 1970), the overall bias B1 and the Root Mean Square Error

(RMSE) are complemented by the temporal bias B2 and the spatial bias B3 (Bárdossy and Pegram, 2012), which are defined as:

$$\rho = \frac{\sum\limits_{i=1}^{n_i} \sum\limits_{j=1}^{J} (Z^*(\mathbf{x}_i,t_j) - \bar{Z}^*) \times (z(\mathbf{x}_i,t_j) - \bar{z})}{\sqrt{\sum\limits_{i=1}^{n_i} \sum\limits_{j=1}^{J} [Z^*(\mathbf{x}_i,t_j) - \bar{Z}^*]^2 \times \sum\limits_{i=1}^{n_i} \sum\limits_{j=1}^{J} [z(\mathbf{x}_i,t_j) - \bar{z}]^2}} \qquad [-] \qquad (17)$$

$$\text{NSE} = 1 - \sum\limits_{i=1}^{n_i} \sum\limits_{j=1}^{J} \frac{[z(\mathbf{x}_i,t_j) - Z^*(\mathbf{x}_i,t_j)]^2}{[z(\mathbf{x}_i,t_j) - \bar{z}]^2} \qquad [-] \qquad (18)$$

$$\text{B1} = \frac{1}{n_{tot}} \sum\limits_{i=1}^{n_i} \sum\limits_{j=1}^{J} [z(\mathbf{x}_i,t_j) - Z^*(\mathbf{x}_i,t_j)] \qquad [\,\text{mm}\,] \qquad (19)$$

$$\text{B2} = \frac{1}{n_{tot}} \sum\limits_{i=1}^{n_i} \left[ \sum\limits_{j=1}^{J} (z(\mathbf{x}_i,t_j) - Z^*(\mathbf{x}_i,t_j)) \right]^2 \qquad [\,\text{mm}^2\,] \qquad (20)$$

$$\text{B3} = \frac{1}{n_{tot}} \sum\limits_{j=1}^{J} \left[ \sum\limits_{i=1}^{n_i} (z(\mathbf{x}_i,t_j) - Z^*(\mathbf{x}_i,t_j)) \right]^2 \qquad [\,\text{mm}^2\,] \qquad (21)$$

$$\text{RMSE} = \sqrt{\frac{1}{n_{tot}} \sum\limits_{i=1}^{n_i} \sum\limits_{j=1}^{J} [z(\mathbf{x}_i,t_j) - Z^*(\mathbf{x}_i,t_j)]^2} \qquad [\,\text{mm}\,] \qquad (22)$$

where $J$ is the number of time steps, $n_i$ is the number of observation locations and $n_{tot}$ is the total number of cross-validated observations. Note that the cross-validation for only one time step ($J = 1$) would yield the following relations: $n_{tot} \times \text{B1}^2 = \text{B3}$ and $\text{B2} = \text{RMSE}^2$.

### 3.2.1 Summary results

The overall values of the six objective functions from all 32226 original observations, along with a separation into dry season (calender months: 4 - 9) and wet season (calender months: 1 - 3 and 10 - 12) are given in Table 1.

TABLE 1

The total values of the correlation coefficient $\rho$, the NSE coefficient, the temporal bias B2 and the RMSE are better for QK-$\Gamma$ and QK-Wei than for OK and EDK, evoking from a superior performance especially during the wet season when not many of many zero-values are present (see Table 1).

Complementary, OK and EDK have superior values for the biases B1 and B3 as a result of the implicit definition as best linear and unbiased estimator. OK (and to some extent EDK) optimize the spatial bias B3 for a given month by adapting their global mean to the observed mean, according to Eq. 1 (Eq. 3). However, this evokes a systematic underestimation in regions of high precipitation and a systematic overestimation in regions of low precipitation. Therefore, a temporal bias B2 accumulates for a

location, which consistently experiences extreme precipitation over time. Especially during the wet season, QK outperforms OK and EDK with respect to the temporal bias. The following investigations on raingauge 'Wilgervier' exemplary serve as illustration for the evolution of a temporal bias.

### 3.2.2 Temporal bias at the raingauge 'Wilgervier'

5 The raingauge 'Wilgervier' (i = 125, see Fig. 3) records a relatively high monthly precipitation of $70.1\,mm$ in average compared to the average monthly precipitation of $54.7\,mm$ in the entire domain.

The evolution of the temporal bias B2 at the raingauge 'Wilgervier' is calculated from cross-validation according to Eq. 20 and illustrated in Fig. 9 (left). In addition, the relative estimation error $\epsilon_r$ is estimated from the $218$ (out of the $264$ possible) original observations at 'Wilgervier', being defined as:

$$\epsilon_r(\mathbf{x}_{125}, t_j) = \frac{Z^*(\mathbf{x}_{125}, t_j) - z(\mathbf{x}_{125}, t_j)}{z(\mathbf{x}_{125}, t_j)} \tag{23}$$

The $218$ values of $\epsilon_r(\mathbf{x}_{125})$ are smoothed by a Gaussian Kernel with a defined range $d_G$ $(= 0.35)$. The distribution of the relative estimation errors should ideally be symmetrical around zero. However, the respective distributions are truncated due the confinement to $\epsilon_r \geq -1$ for the variable of monthly precipitation. The smoothed distributions are factually a summary of the estimation errors and are illustrated in Fig. 9 (right).

FIGURE 9

OK displays the highest systematic underestimation over time (Fig. 9 (left)) and the relative estimation errors have a mode of $-20\,\%$ (Fig. 9 (right)). EDK slightly improves the systematic bias of the interpolation, but the relative error distribution still possess a mode of $-15\,\%$. QK-$\Gamma$ and QK-Wei can further improve the systematic underestimation over time and exhibit error distributions with modes of $-12\,\%$, and $-10\,\%$ respectively.

The raingauge 'Wilgervier' illustrates that OK and EDK might optimize the spatial bias (Table 1), but they are hampered to minimize the temporal bias in locations of extreme observations. QK, as a spatio-temporal interpolation method, is capable of reducing the temporal bias within regions of relatively high (or low) precipitation, which is potentially important for possible successive water balance considerations.

### 3.2.3 Cross-validation for different calendar months

The effects of the increased occurrence of zero-value observations on the Pearson correlation coefficient $\rho$ (Eq. 17) and the RMSE (Eq. 22) is exemplary examined next. The respective values are calculated for each calender month from the cross-validation and are illustrated in Fig. 10 along with the dry ratio (Fig. 4).

FIGURE 10

QK-Γ and QK-Wei display improved values in comparison to OK and EDK for the two selected objective functions from October to March (Fig. 10). However, their performance deteriorates from May to September, when many zero-value observations are present, indicated by a dry ratio of at least $25\%$ or above. The correlation coefficient $\rho$ plunges in July for both versions of QK (Fig. 10 (left)) and the respective RMSE shows a similar qualitative behavior (Fig. 10 (right)).

The performance of QK is considerably influenced by the dry ratio. The presence of many zero-values in the data leads to very steep or nearly vertical theoretical cdfs, hampering the allocation of the quantiles to the respective precipitation values.

### 3.3 Cross-validation of the uncertainty

The estimated error distribution of the estimator $Z^*(\mathbf{x})$ is described by the associated standard deviation $\sigma_K(\mathbf{x})$ as a measure of associated uncertainty. The quality of the uncertainty from the cross-validation is assessed by two objective functions: the

10 adapted Linear Error in the Probability Space LEPS (Ward and Folland, 1991) and a test on uniformity (Bárdossy and Li, 2008).

LEPS compares the values of the estimator $Z^*(\mathbf{x}_i, t)$ and the observation $z(\mathbf{x}_i, t)$ within the estimated cdf $F_{Z^*}$ of the error distribution as:

$$\text{LEPS} = \frac{1}{n_{tot}} \cdot \sum_{i=1}^{n_{tot}} |F_{Z^*}(z(\mathbf{x}_i, t)) - F_{Z^*}(Z^*(\mathbf{x}_i, t))| \tag{24}$$

LEPS is defined on the interval $[0, 1]$: low values indicate a higher probability for the observation to originate from the estimated probability density distribution and v.v.. The average over the differences of all observations $n_{tot}$ yields the overall LEPS value.

The test on uniformity verifies the estimated, conditional distribution $F_{Z^*}$ by calculating its value $F_{Z^*}(z(\mathbf{x}_i, t))$ for every original observation $z(\mathbf{x}_i, t)$. The resulting values (or quantiles) should be uniformly distributed on the interval $[0, 1]$

(Bárdossy and Li, 2008). We defined ten classes of equal width, which should have the same resulting relative frequency. The deviation from uniformity is quantified by the $\chi^2$- test variable as the sum of the relative squared differences between uniformity and empirical distribution, ranging from zero (perfect) to nine (improper).

### 3.3.1 Summary results

The values of the two objective functions from cross-validation of all 32226 original observations of the entire year, and divided

into dry (calender months: 4 - 9) and wet season (calender months: 1 - 3 and 10 - 12) are displayed in Table 2.

TABLE 2

The best overall LEPS values are received from the traditional EDK and OK (Table 2). QK-Wei is superior to QK-Γ, but both versions of QK are displaying higher LEPS values than OK or EDK, originating from the dry season when many zero-values are present in the data.

However, the $\chi^2$-test variables (Table 2) exhibit a reverse hierarchy among the implemented interpolation methods: QK is superior during the dry season and similiar during the wet season. The $\chi^2$-test variables should be read in conjunction with the corresponding histograms of the $F_{Z^*}$ values (Fig. 11). Note that the outer classes of the histograms host all the observations $z(\mathbf{x}_i, t)$, which are situated outside the estimated distribution. These classes exhibit the largest deviation from the ideal uniform distribution.

## FIGURE 11

QK-$\Gamma$ and QK-Wei provide in general a more uniform distribution of $F_{Z^*}$ than OK and EDK (Table 2), which possess the same value of the $\chi^2$-test variable and similar histograms (Table 2 and Fig. 11) due to their implicit affinity in definition.

### 3.3.2 Cross-validation for different calendar months

The effect of many zero-value observations on the error distribution is investigated by the differentiation into calendar months. The objective functions are recalculated accordingly and illustrated in Fig. 12.

## FIGURE 12

The temporal evolution of the LEPS values for the two versions of QK is influenced by the presence of many zero-value observations. QK-$\Gamma$ and QK-Wei exhibit LEPS values superior to OK and EDK from September to April, characterized by a dry ratio of less than $26\%$ (Fig. 12 (left)). However, the performance of QK deteriorates from May to August when many zero-value observations are present. This dependence explains the overall inferior LEPS values for QK in Table 2. The LEPS values for OK and EDK are hardly influenced by the dry ratio (Fig. 12 (left)) and show a congruent behavior.

The temporal evolution of the $\chi^2$ test variable (Fig. 12 (right)) shows better values for QK-$\Gamma$ and QK-Wei than for OK or EDK during most calender months. QK maintains a more uniform distribution of the $F_{Z^*}$ values even for the months with a high dry ratio when OK and EDK deteriorate.

The cross-validation for the uncertainty suggests an improvement by QK under the prerequisite of a low dry ratio within the input data. This improvement is attributed to the wider range of the error distribution and the increased relation between the magnitude of the estimator and the spread of the distribution (see Section 3.1).

## 4 Conclusions

The geostatistical interpolation method of QK addresses the spatial non-stationarity of a variable of interest by its conversion into quantiles and defining distribution parameters. The spatial-temporal description of the variable by QK is a novelty in applied geostatistics and can be regarded as a temporal extension of Probability Kriging. Therefore, the proposed method could be extended to spatially aggregated variables of streamflows, requiring, however, further investigations. The proposed method accommodates skewed marginal distributions and converts them into an ideal Gaussian distribution prior to interpolation as a major theoretical advantage over the traditional OK or EDK. QK describes an asymmetrical distribution of the random

variable $Z(\mathbf{x})$ by the non-linear estimator $Z^*(\mathbf{x})$ and the estimation variance $\sigma_K^2(\mathbf{x})$ of the error. QK further establishes a relation between the magnitude of both descriptors.

The variable of monthly precipitation, observed at $226$ raingauges over $264$ consecutive time steps, serves as input data. We selected the two parametric $\Gamma$- distribution and Weibull distribution, because they are defined on the interval $[0, \infty]$ and are suitable to describe the variable of monthly precipitation. The selected distributions are fitted to the observations of a specific calendar month, implying an absence of temporal dependence between two sample members (e.g. between the monthly precipitation of December 2002 and December 2003). However, QK does accommodate temporal independence between consecutive observations, unlike existing spatio-temporal Kriging methods. In general, other types of distributions, with a higher number of parameters could be selected, especially in case of other variables of interest. Finally, we used elevation as external drift, both for the interpolation of the parameters within QK as well as for the reference EDK.

The cross-validation of the estimator revealed an improvement for most of the selected objective functions. In particular, QK addresses the temporal bias, which remains unattended by the traditional geostatistical methods, which only optimize the mean spatial bias. In case of the estimator, QK-$\Gamma$ performs slightly better than QK-Wei for most of the selected objective functions. The cross-validation of the associated uncertainty shows an improvement by QK in the description of the distribution of the estimation errors in comparison to the traditional geostatistical interpolation methods. However, its performance depends on the percentage of zero-values in the input data and declines when many zero-values are present. In general, QK-Wei shows a superior estimation of the associated uncertainty than QK-$\Gamma$.

*Code availability.* Respective codes can be obtained from the corresponding author.

*Data availability.* Precipitation and elevation data can be obtained from the respective sources mentioned in Sect. 2.3.1

*Competing interests.* The authors declare that they have no conflict of interest.

*Acknowledgements.* This research was executed at the Institute for Modeling Hydraulic and Environmental Systems of the University of Stuttgart.

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

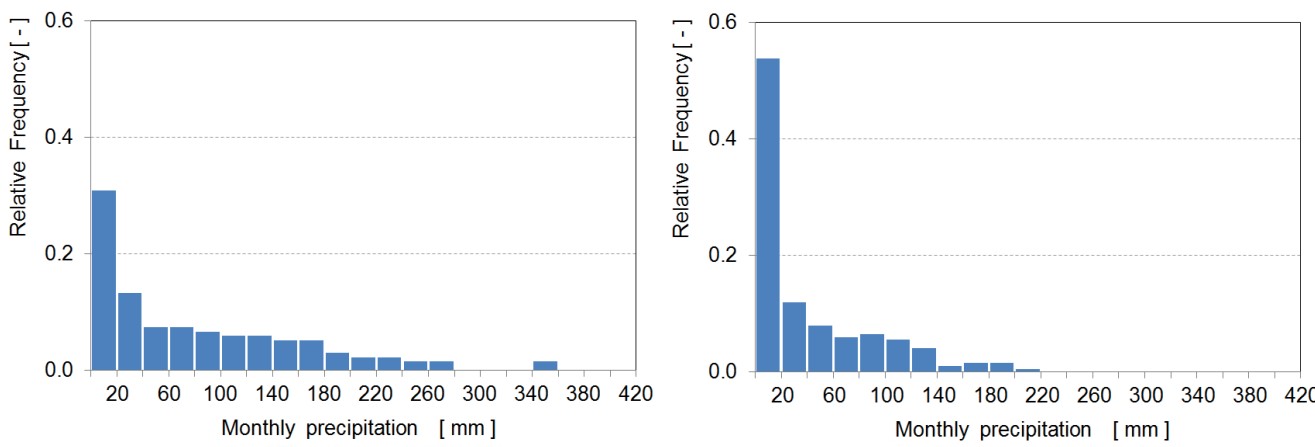

**Figure 1.** Histogram from the times series of observed monthly precipitation for two random stations: 'Laingsnek', $\mathbf{x}_{57}$ with $\overline{z} = 81.3\,mm$ (left) and 'Tambotieboom', $\mathbf{x}_{29}$ with $\overline{z} = 38.3\,mm$ (right).

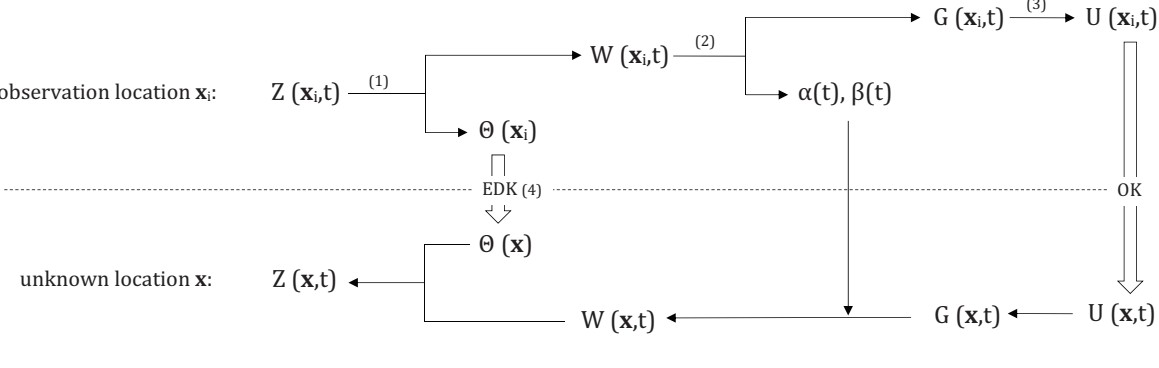

(1)  Fitting of temporal distribution F(z;Θ)     (3)  Normal-Score Transformation into U[0,1]

(2)  Transformation into Beta- distribution G(w;α,β)     (4)  incl.  PCA

**Figure 2.** Flowchart for the basic methdology of Quantile Kriging.

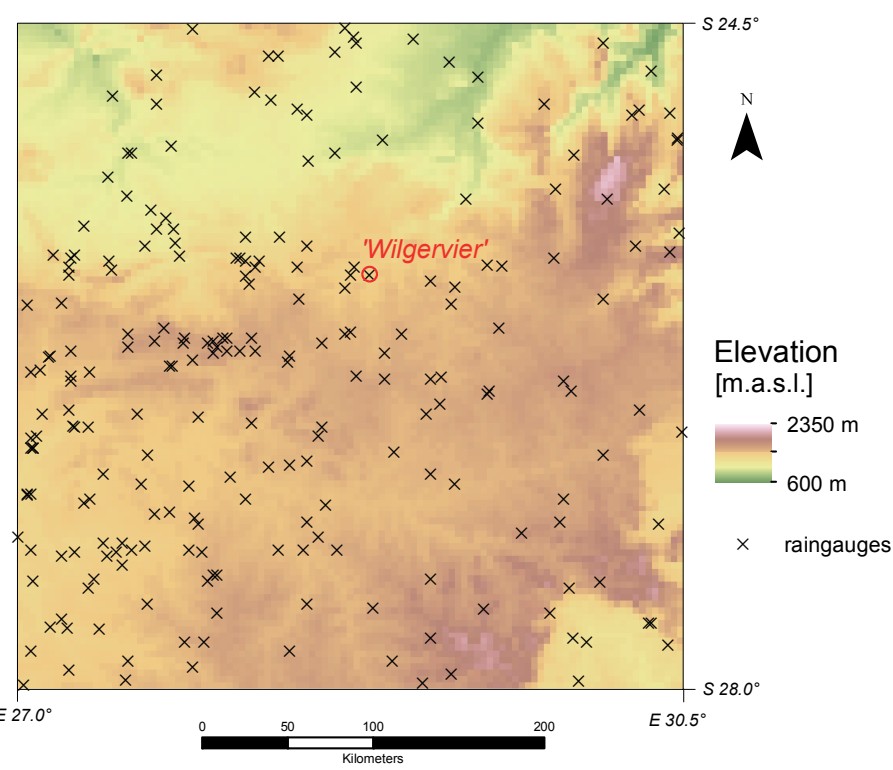

**Figure 3.** Study area, elevation and location of raingauges.

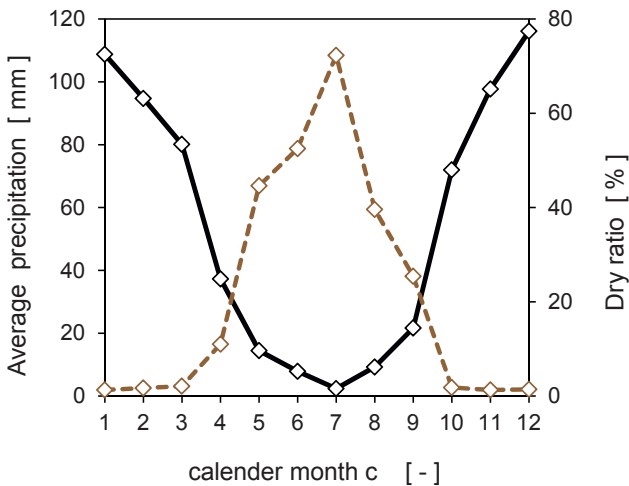

**Figure 4.** Average monthly precipitation (in mm) and dry ratio (in %) from 226 raingauges. Note that the dry ratio (dashed brown line) is indicated on the right axis.

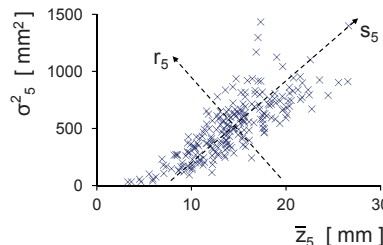

**Figure 5.** Scatterplot of sample mean $\bar{z}_5$ and sample variance $\sigma_5^2$ of calendar month 'May' along with the principle components $s_5$ and $r_5$.

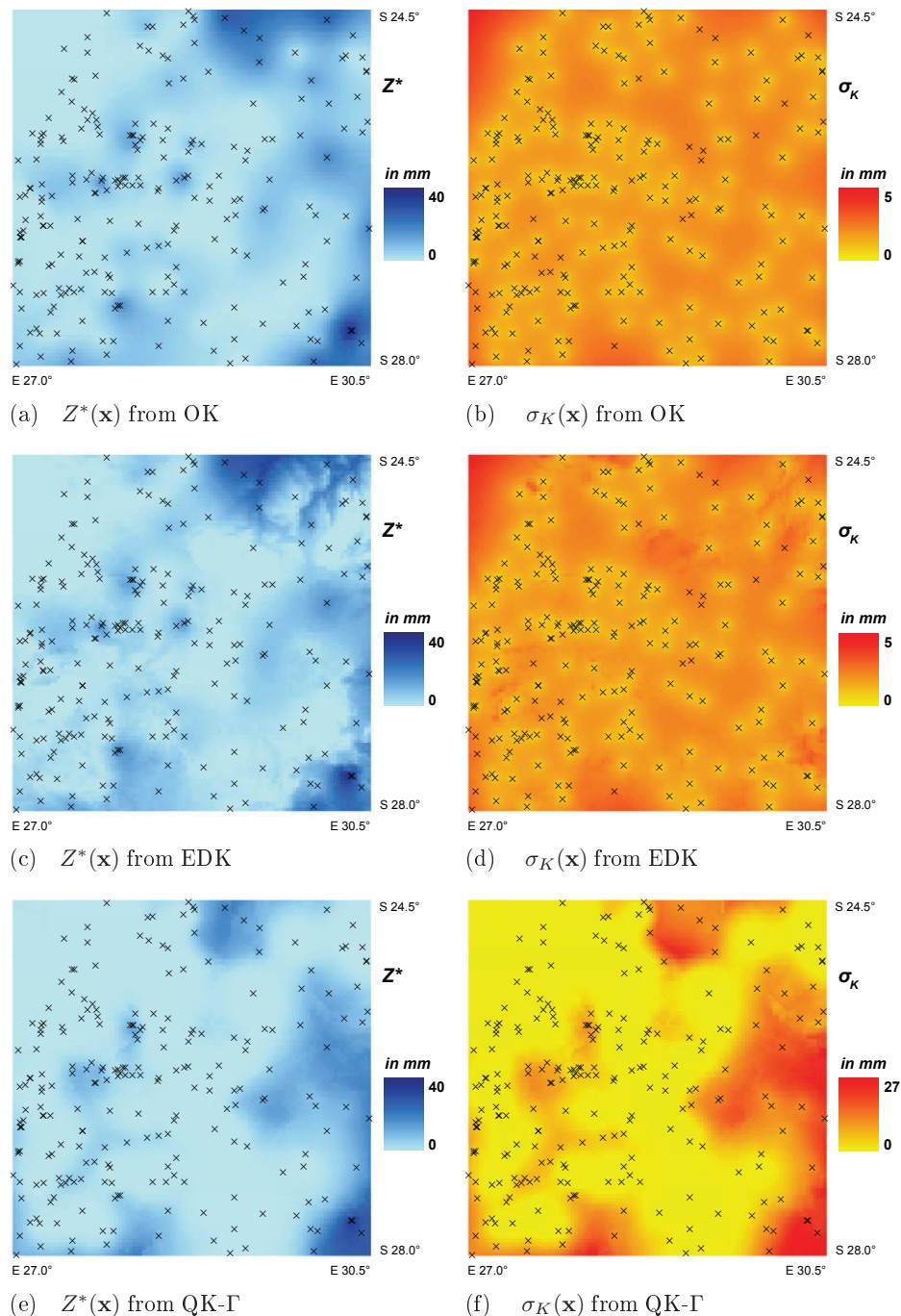

(a)  $Z^*(\mathbf{x})$ from OK

(b)  $\sigma_K(\mathbf{x})$ from OK

(c)  $Z^*(\mathbf{x})$ from EDK

(d)  $\sigma_K(\mathbf{x})$ from EDK

(e)  $Z^*(\mathbf{x})$ from QK-$\Gamma$

(f)  $\sigma_K(\mathbf{x})$ from QK-$\Gamma$

**Figure 6.** Spatial patterns of the estimator $Z^*(\mathbf{x})$ and the standard deviation $\sigma_K(\mathbf{x})$ from OK, EDK and QK-$\Gamma$ for August 1993. Note that crosses indicate positions of raingauges.

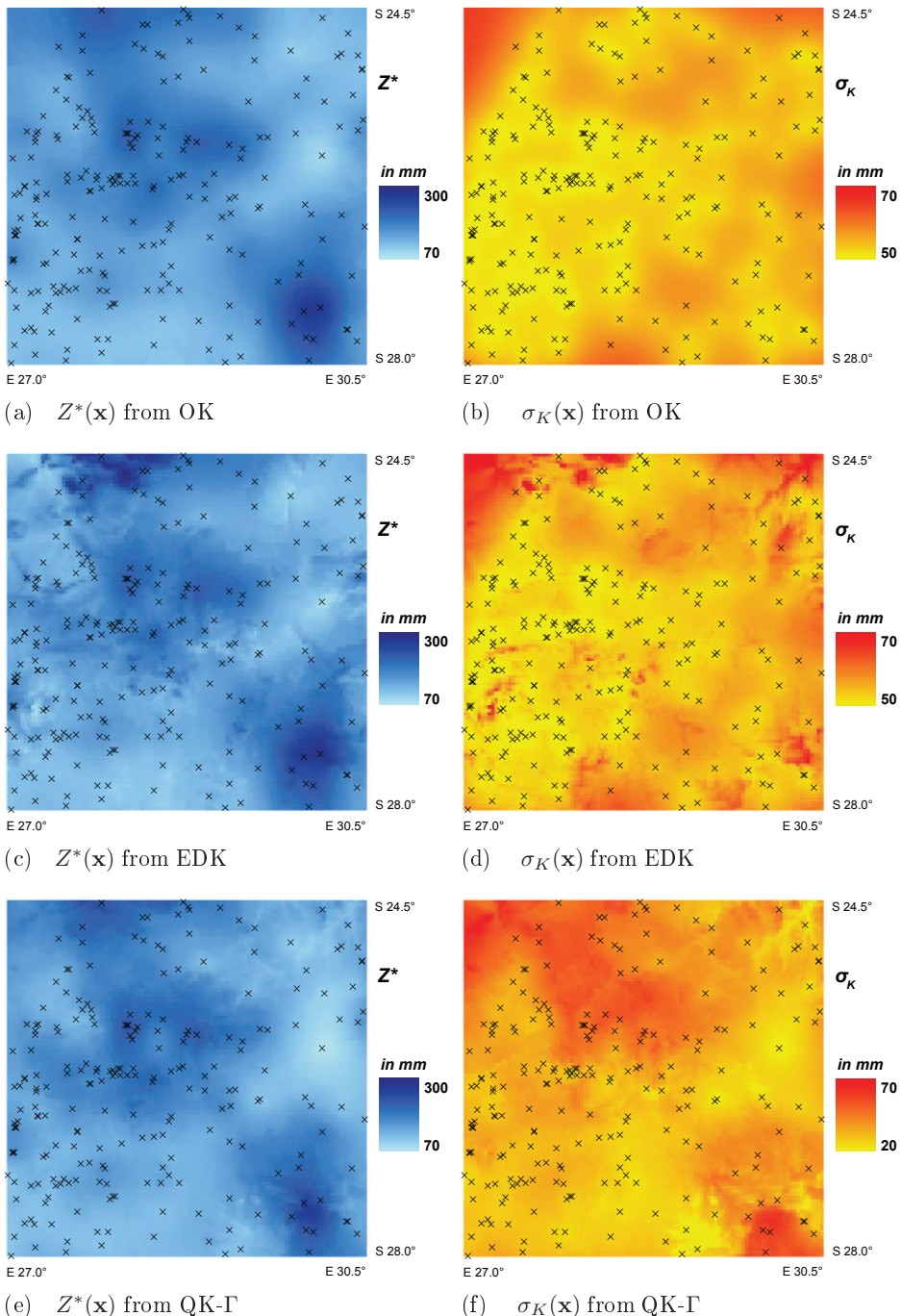

**Figure 7.** Spatial patterns of the estimator $Z^*(\mathbf{x})$ and the standard deviation $\sigma_K(\mathbf{x})$ from OK, EDK and QK-$\Gamma$ for January 1996. Note that crosses indicate positions of raingauges.

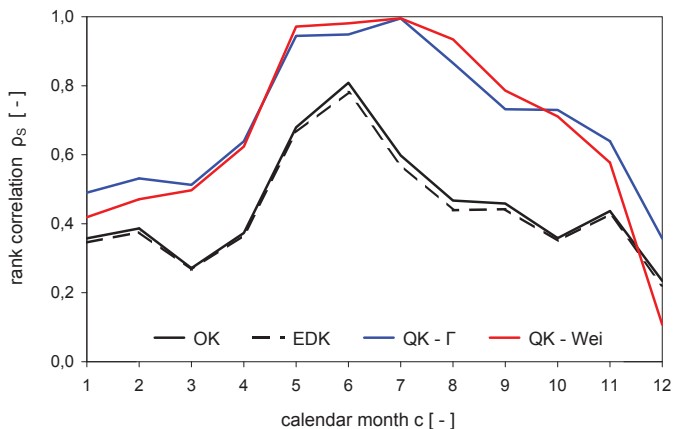

**Figure 8.** Evolution of the Spearman rank correlation coefficient $\rho_S$ between the estimator $Z^*$ and the standard deviation $\sigma_K$.

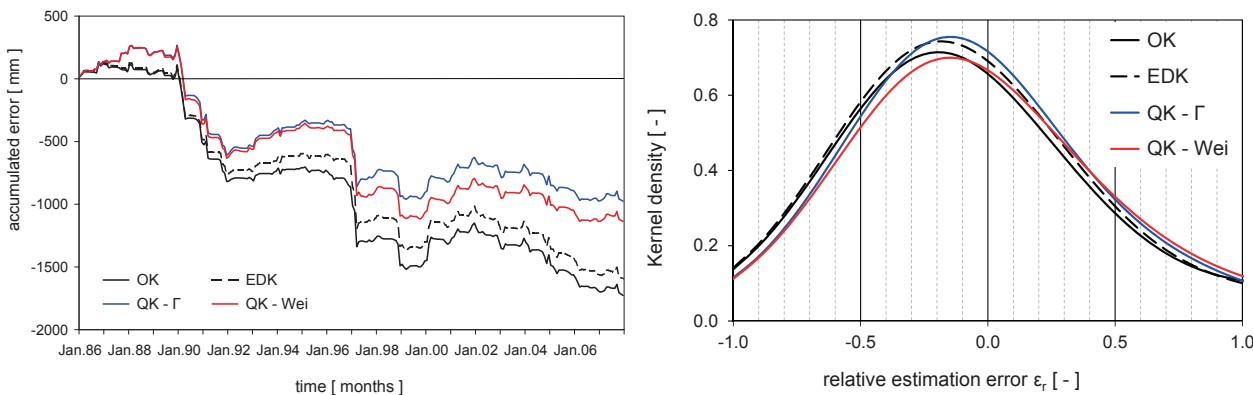

**Figure 9.** Errors of the estimator $Z^*(\mathbf{x}_{125})$ at raingauge 'Wilgervier': Evolution of temporal bias B2 over the study period (left) and Smoothed distribution of the relative estimation error $\epsilon_r$ (right).

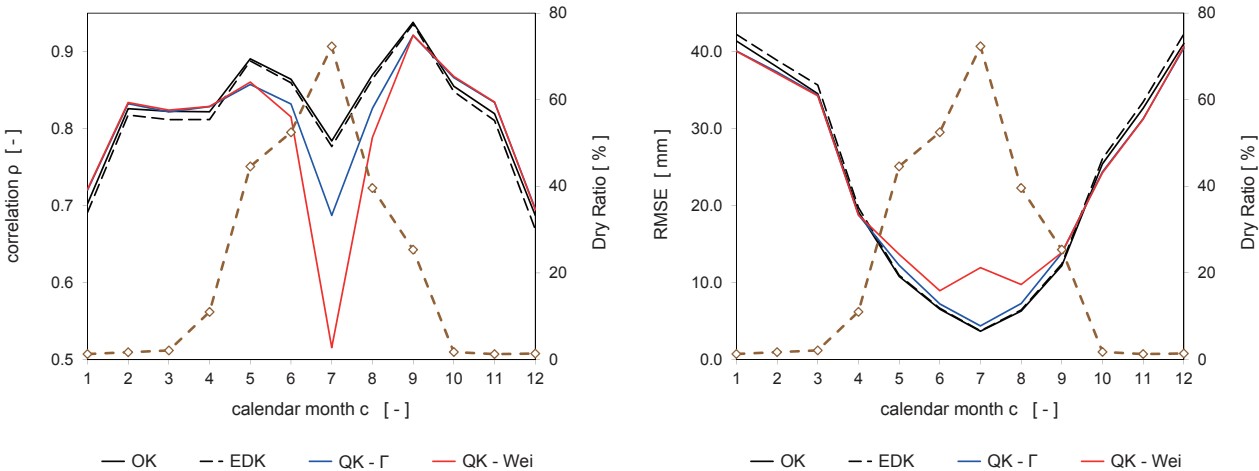

**Figure 10.** Evolution of two objective functions for the estimator over the twelve calendar months: Correlation coefficient $\rho$ (left) and RMSE (right). Note that the dry ratio (dashed brown line) is indicated as percentage on the right axis.

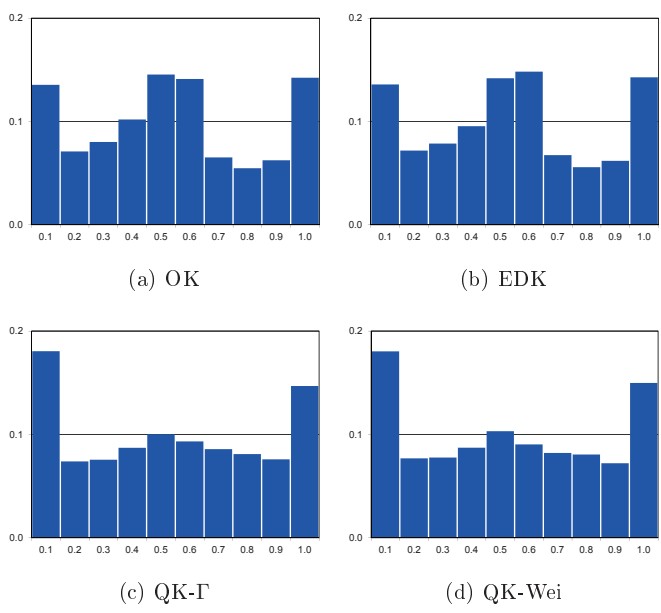

**Figure 11.** Histograms for the $F_{Z^*}$-values of four different interpolation methods.

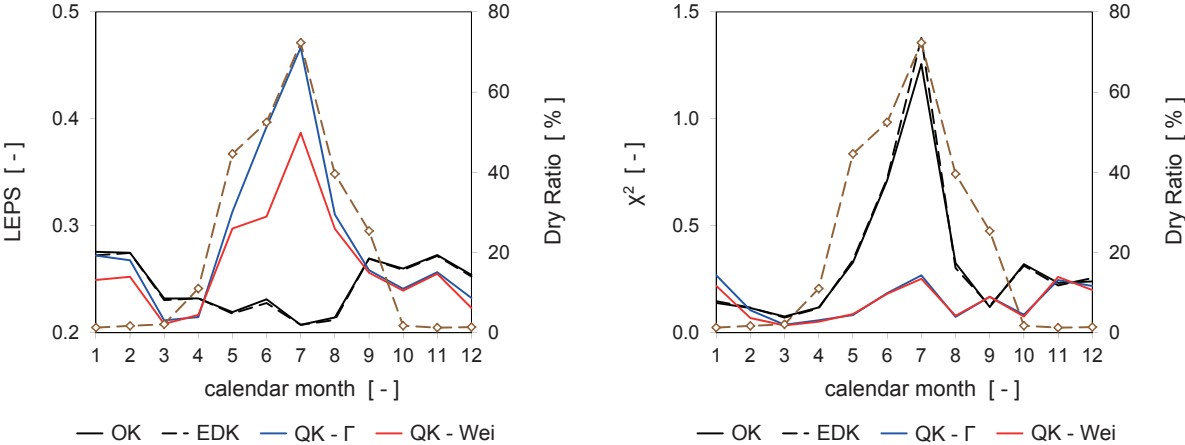

**Figure 12.** Evolution of the two objective functions for the error distribution over twelve calendar months: LEPS (left) and $\chi^2$ (right). Note that the dry ratio (dashed brown line) is indicated as percentage on the right axis.

**Table 1.** Summary results from the cross-validation of the estimator $Z^*$ for the twelve calender months of the entire year, and spit into dry (calender months: 4 - 9) and wet (calender months: 1 - 3 and 10 - 12) season.

| | $n_{tot}$ | $\rho$ | NSE | B1 | B2 | B3 | RMSE |
|---|---|---|---|---|---|---|---|
| | $[-]$ | $[-]$ | $[-]$ | $[mm]$ | $[mm^2]$ | $[mm^2]$ | $[mm]$ |
| *Entire year:* | | | | | | | |
| OK: | 32226 | 0.902 | 0.813 | 0.07 | 3486.34 | 49.67 | 26.48 |
| EDK: | 32226 | 0.897 | 0.803 | 0.02 | 3539.73 | 76.41 | 27.16 |
| QK-$\Gamma$: | 32226 | 0.905 | 0.819 | -0.45 | 3480.90 | 1346.32 | 26.06 |
| QK-Wei: | 32226 | 0.903 | 0.814 | 0.75 | 3418.60 | 2105.09 | 26.40 |
| *Dry season:* | | | | | | | |
| OK: | 16256 | 0.908 | 0.824 | 0.00 | 2663.82 | 11.42 | 11.05 |
| EDK: | 16256 | 0.904 | 0.816 | 0.01 | 2731.59 | 15.82 | 11.30 |
| QK-$\Gamma$: | 16256 | 0.897 | 0.803 | 0.46 | 2164.02 | 1147.99 | 11.69 |
| QK-Wei: | 16256 | 0.875 | 0.748 | 2.38 | 2153.02 | 3323.72 | 13.24 |
| *Wet season:* | | | | | | | |
| OK: | 15970 | 0.801 | 0.637 | 0.14 | 2939.87 | 88.60 | 35.93 |
| EDK: | 15970 | 0.790 | 0.618 | 0.02 | 3034.41 | 138.09 | 36.86 |
| QK-$\Gamma$: | 15970 | 0.809 | 0.654 | -1.39 | 2509.63 | 1548.21 | 35.10 |
| QK-Wei: | 15970 | 0.810 | 0.654 | -0.91 | 2523.36 | 864.63 | 35.05 |

**Table 2.** Summary results from the cross-validation of the estimation error for the entire year, and split into dry (calender months: 4 - 9) and wet (calender months: 1 - 3 and 10 - 12) season.

| | $n_{tot}$ [$-$] | LEPS [$-$] | $\chi^2$ [$-$] | $n_{tot}$ [$-$] | LEPS [$-$] | $\chi^2$ [$-$] | $n_{tot}$ [$-$] | LEPS [$-$] | $\chi^2$ [$-$] |
|---|---|---|---|---|---|---|---|---|---|
| | *Entire year:* | | | *Dry Season:* | | | *Wet Season:* | | |
| OK: | 32226 | 0.25 | 0.13 | 16256 | 0.19 | 0.32 | 15970 | 0.30 | 0.17 |
| EDK: | 32226 | 0.24 | 0.13 | 16256 | 0.19 | 0.33 | 15970 | 0.30 | 0.17 |
| QK-$\Gamma$: | 32226 | 0.32 | 0.11 | 16256 | 0.36 | 0.10 | 15970 | 0.28 | 0.18 |
| QK-Wei: | 32226 | 0.26 | 0.12 | 16256 | 0.27 | 0.10 | 15970 | 0.25 | 0.18 |