# Peer review of "Geostatistical interpolation by Quantile Kriging"

_Hydrology and Earth System Sciences, 2018_

## Referee Comment (RC1) · Anonymous Referee #1 · 18 Jun 2018

**Review of the manuscript titled "Geostatistical Interpolation by Quantile Kriging" by H. Lebrenz and A. Bardossy**

The proposed manuscript presents a new geo-statistical interpolation method (Quantile Kriging – QK) that is able to relax three of the main assumption/limitations of the most used Ordinary Kriging: 1) spatial stationarity of the process mean, 2) Gaussianity of the interpolated variable and 3) independence of the uncertainty on the estimation value. The work extends the formulations of other well-known kriging methods with logic and statistical rigour. Although the presented technique still has a major limitation in the ability to handle the presence of many zero values (as often happens when dealing with rainfall, especially at finer scales than the presented one), it can be considered an improvement on the state of the art and a contribution to the advancement of the field. Additionally, although the authors do not mention it in the manuscript (and should) there are many applications to a variety of environmental variables where the presence of zeros is not a problem and the presented technique could be better applied. The manuscript is very well written and easy to follow. I suggest the following improvements:

1. The introduction explains a lot about the evolution of kriging techniques. However a little bit more discussion about applications (especially to rainfall) their limitations in hydrology, the main challenges, etc… could help defining the framework.
2. P.4 l. 18, many of the presented geostatistical techniques were developed in geological sciences, where the temporal evolution of the studied variables is often irrelevant. I would mention this to explain why the temporal variability is often ignored in kriging.
3. I would mention spatio-temporal kriging and other similar techniques as attempts to incorporate the temporal variability. How is this method better/different (e.g. Gaussianity)? Examples:
    - Snepvangers, J. J. J. C., Heuvelink, G. B. M., & Huisman, J. A. (2003). Soil water content interpolation using spatio-temporal kriging with external drift. Geoderma, 112, 253–271. https://doi.org/10.1016/S0016-7061(02)00310-5
    - Sideris, I. V., Gabella, M., Erdin, R., & Germann, U. (2014). Real-time radar-rain-gauge merging using spatio-temporal co-kriging with external drift in the alpine terrain of Switzerland. Quarterly Journal of the Royal Meteorological Society, 140(April), 1097–1111. https://doi.org/10.1002/qj.2188
4. Eq. 5: I am not sure why you fit a Beta distribution to the quantiles: isn't the Normal Score Transformation (NST) designed to work with empirical distributions?
5. Eq.6 and Eq.7: You applied the NST, so isn't this $E[F(U)] = m$ and same for Variance? Maybe I'm missing something
6. Nowhere is explained how you calculate the variograms for all the interpolations you do. Maybe worth mentioning it somewhere.
7. Pg 7 top: you introduce the elevation dataset, but you don't explain why. Mention you use it for both EDK of the parameters and for the reference EDK of the rainfall process.
8. is the dry ratio the number of stations that recorded zero rainfall over the whole month over the total number of stations? Can you state this a bit more explicitly?

9. P.7, l. 18: if you fit a PDF for each month for each station, you have only 22 points to do it, it seems a very little number to be statistically robust. maybe one of the reasons why you need to fit mean and variance rather than the parameters?

10. Eq. 8 and eq. 9: you here present both the distributions but don't explain why. Do you want to compare their performance? How did you select Gamma and Weibull distributions? Nowhere in the paper you comment on which one performs best overall.

11. P.8, l.27: You need to state that you do EDK with elevation as the drift. One of the problems I have in this comparison is that often EDK is performed with radar data, which probably would do better than elevation in defining the spatial pattern of rainfall. Can you comment on this?

12. P.9, l. 19: One of the drawbacks I observe is that QK does not estimate a higher uncertainty where there are less rain gauges, eg. top left corner of Figure 5f.

13. Explain what rho (eq. 12) represent, why you use it, what is its range, and what the optimal value)

14. P 13: I find the explanation about chi squared a bit confusing. I could not understand what had to be uniform and why, until later on you introduce the histogram. Maybe worth introducing the histograms first? or at least explain more in details.

15. Conclusions: You need to write more here, and remove one of the two paragraphs that are repeated (l. 20-26 or 27-3).

16. I feel in general a little bit more discussion of the overall results could be introduced either in the Results and Discussion or the Conclusion section, including many of the comments I mentioned before.

---

## Referee Comment (RC2) · Anonymous Referee #2 · 27 Jun 2018

The authors propose a new kriging technique developed to regionalize non-normally distributed spatio-temporal variables. The approach, as I understand it, involves (i) fitting time series observations to a predetermined distribution type at each gauge independantly; (ii) using the fitted distribution to assign a probability of non-exceedance to each observation at each gauge; (iii) fit the subset of these probabilities corresponding to each observation time to (different) beta distributions; (iv) map the fitted probabilities to normal quantiles; (v) use the (now normally distributed and *assumed* second-order stationary) outcomes as a basis to apply OK or EDK, as described (rather cryptically) in section 2.1.2.

While the approach is intriguing, its description in the paper lacks statistical rigor and minimal proofs and intuitions. Its application in cross validation suggests that it does

a decent job at predicting some measure of rainfall, but so would most properly fitted interpolation methods. This does not mean that the estimator is BLUE (best – or even efficient – unbiased linear estimator) of the considered stochastic process. Actually, the apparent correlation between Z*(x) and sigma2(x) (Figure 4) suggests that the process has some degree of heteroskedasticity. Under these conditions, linear models (like most kriging estimators) are not necessarily efficient (see, for instance, the Gauss Markov Theorem for ordinary least squares). Again, the approach is promising, but its exposition needs a major overhaul to be convincing.

In particular:

Non-Gaussianity: In its canonical form, Ordinary Kriging is based on the method of moment (i.e. variance minimization subject to unbiasedness) and so is not technically restricted to normally distributed processes. Gaussiannity is, however, required for maximum likelihood (ML) type estimations, which some studies have shown can be more efficient for kriging-based regionalization (e.g., Lark 2000). ML approaches become necessary to have unbiased prediction of both the mean and the variance when it comes to EDK or Universal Kriging, particularly when subject to more intricate error correlation structures (e.g., Muller 2015). Since non-gaussianity appears to be a key rationale for the proposed approach, it is important to be specific on that.

Beta-distribution: The use of the beta distribution is intriguing, but more intuition is needed on the assumed underlaying stochastic process. Please describe clearly the properties of the (space-time) stochastic process that you assume gives rise to the observed sample and use that to demonstrate Eqn 6 and 7 in a rigorous mathematical proof. I find the use of the beta distribution promising because a common interpretation is that it describes the distribution of the probabilities associated to a binomial process observed over a finite sample. Let's say that the binomial process in question is the exceedance of a given threshold (as eluded to in the manuscript). Then, if the underlaying point process is identically distributed in space and if an identically sized sample is taken at each observation point, the proportion of observations lower than a

given threshold across all gauges will be beta-distributed. Perhaps that's a start?

Stationarity: More fundamentally, a main issue that I have is that your approach involves fitting distributions independantly at different points in space (gamma or weibull) and time (beta), which implies that the underlaying random point process follows a different distribution at each point in space. Granted, EDK and Universal kriging allow the first moment of the underlaying distribution to vary through space, as allowed by the scaling properties of the expected value estimator. However, I am not aware of any existing geostatistical approach that allows for higher order moments to vary through space. This may be fine, but please demonstrate that your approach does not violate the second order stationarity assumption (i.e that the variogram is constant through space), which is critical (and arguably more important than gaussianity) for kriging.

References:

Lark, R.M. (2000) "Estimating variograms of soil properties by the method of moments and maximum likelihood", European Jornal of Soil Science

Muller, M.F and Thompson, S.E. (2015), "TopREML: a topological restricted maximum likelihood approach to regionalize trended runoff signatures in stream networks", HESS

---

## Referee Comment (RC3) · Anonymous Referee #3 · 2 Jul 2018

The article presents an interesting approach to kriging with skewed variables and to non-stationarity: i) For every single location the distribution over time is estimated and quantiles are estimated. ii) To the quantiles of a given time-step a Beta-distribution is fitted. iii) The quantiles of the Beta-distribution are transformed by a Normal-Score transformation into standard Gaussian variables. iv) Ordinary kriging of the transformed variables. v) Backtransformation of the kriging results to the original scale

One to my opinion main result now is that the variance of the prediction is dependent on the data values themselves, too, and not as in ordinary kriging only dependent on the kriging location. The methodology reminds me somewhat to trans-Gaussian kriging, where you have a similar effect, with the difference that you are still stationary. Maybe you could a little bit comment on this and also on the relationship to copulas.

[Figure]

Non-stationarity comes into play because you estimate at each spatial location the quantiles separately. You calculate quantiles, and quantiles are always related to copulas,- is there also here a relationship to copulas? Please, elaborate on that. I am also not completely sure, why you need the Beta-distribution at all and not directly calculate the Normal-Score transformation.

---

## Author Comment (AC1) · 14 Aug 2018

**Responses on the Referees 1 comments on the submitted manuscript "Geostatistical interpolation by Quantile Kriging" hess-2018-276**

We are very thankful to the anonymous referees for their remarks on our submitted manuscript. We believe that they will significantly improve the quality of the manuscript.

We use the nomination e.g. A1.13 (i.e. A(nswer)1 (no. of reviewer).13 (no. of comment)) and numbers (page, line, figures, tables) of the original manuscript submitted in order to address all queries raised:

**RC 1:**

The proposed manuscript presents a new geo-statistical interpolation method (Quantile Kriging – QK) that is able to relax three of the main assumption/limitations of the most used Ordinary Kriging: 1) spatial stationarity of the process mean, 2) Gaussianity of the interpolated variable and 3) independence of the uncertainty on the estimation value. The work extends the formulations of other well-known kriging methods with logic and statistical rigour. Although the presented technique still has a major limitation in the ability to handle the presence of many zero values (as often happens when dealing with rainfall, especially at finer scales than the presented one), it can be considered an improvement on the state of the art and a contribution to the advancement of the field. Additionally, although the authors do not mention it in the manuscript (and should) there are many applications to a variety of environmental variables where the presence of zeros is not a problem and the presented technique could be better applied. The manuscript is very well written and easy to follow. I suggest the following improvements:

1.   The introduction explains a lot about the evolution of kriging techniques. However a little bit more discussion about applications (especially to rainfall) their limitations in hydrology, the main challenges, etc… could help defining the framework.

     A1.1: see A1.3

2.   P.4 l. 18, many of the presented geostatistical techniques were developed in geological sciences, where the temporal evolution of the studied variables is often irrelevant. I would mention this to explain why the temporal variability is often ignored in kriging.

     A1.2: see A1.3

3.   I would mention spatio-temporal kriging and other similar techniques as attempts to incorporate the temporal variability. How is this method better/different (e.g. Gaussianity)? Examples:

     •   Snepvangers, J. J. J. C., Heuvelink, G. B. M., & Huisman, J. A. (2003). Soil water content interpolation using spatio-temporal kriging with external drift. Geoderma, 112, 253–271. https://doi.org/10.1016/S0016-7061(02)00310-5
     •   Sideris, I. V., Gabella, M., Erdin, R., & Germann, U. (2014). Real-time radar-rain-gauge merging using spatio-temporal co-kriging with external drift in the alpine terrain of Switzerland. Quarterly Journal of the Royal Meteorological Society, 140(April), 1097–1111. https://doi.org/10.1002/qj.2188

     A1.3: We try to address all three comments (i.e. A1.1, A1.2 and A1.3) by rephrasing and extending the existing paragraph by the following, starting at p.4, l.18:

     *[…]. The inclusion of a temporal behavior into the geostatistic models is mostly irrelevant for the original geological variables. However, the temporal variability of a variable becomes more prominent for other sciences, e.g. hydrology, where observations from raingauges over several time steps are implemented into the geostatistical models in order to generate spatial precipitation estimates. These estimates subsequently serve as input to the hydrological modelling (e.g. Syed et al. 2003; Basistha et al. 2008; Cole et al. 2008) over multiple time steps. Associated errors in the precipitation estimates may ultimately lead to greater errors in the subsequent discharge modelling (Kobold et al. 2005). These errors*

*strongly depend on the spatial and temporal distribution of the input precipitation (Gabellani et al. 2007, Moulin et al. 2009) and may limit the accuracy of rainfall-runoff simulations.*

*There are geostatistical space-time models in order to incorporate the temporal variability of the variable, but they are primarily aiming on the extrapolation of the variable in time (Snepvangers et al. 2003). Therefore, they require a strong dependence of the variable over time, suited e.g. for groundwater modeling where temporal changes occur relative slowly. This temporal dependence might be absent for other variables, e.g. monthly precipitation. […]*

For demonstration purposes, we remove the seasonality from our observations by subtracting the mean precipitation of the specific calendar month from the observed precipitation. The graph below shows the autocorrelation (with lag 1) for all 226 raingauges (see graph below). There is hardly any (linear) dependence between the monthly precipitation of two successive months, i.e. if a specific month becomes "wet" or "dry" does hardly depend on if the preceding month was relative "wet" or "dry".

[Figure]

*References:*

- Syed, K.H., Goodrich, D.C.. & Myers, D. E. (2003). Spatial characteristics of thunderstorm rainfall fields and their relation to runoff. Journal of Hydrology, 271, 1–21. https://doi.org/10.1016/S0022-1694(02)00311-6

- Basistha, A., Arya, D.S.. & Goel, N. K. (2008). Spatial Distribution of Rainfall in Indian Himalayas -- A Case Study of Uttarakhand Region. Water resources Management, 22, 1325–1346. http://dx.doi.org/10.1007/s11269-007-9228-2

- Cole, S.J., Moore, R.J. (2008). Hydrological modelling using raingauge- and radar-based estimators of areal rainfall. Journal of Hydrology, 358, 159 - 181. https://doi.org/10.1016/j.jhydrol.2008.05.025

- Kobold, M., Suselj, K. (2005). Precipitation forecasts and their uncertainty as input into hydrological models. Hydrology and Earth System Sciences, 9, 322-332. https://www.hydrol-earth-syst-sci.net/9/322/2005/

- Gabellani, S., Boni, G. Ferraris, L., von Hardenberg J., & Provenzale A. (2007). Propagation of uncertainty from rainfall to runoff: A case study with a stochastic rainfall generator. Advances in Water Resources, 30, 2061 - 2071. https://doi.org/10.1016/j.advwatres.2006.11.015

- Moulin, L., Gaume, E. & Obled C. (2009). Uncertainties on mean areal precipitation: assessment and impact on streamflow simulations. Hydrology and Earth System Sciences, 13, 99-114. https://www.hydrol-earth-syst-sci.net/13/99/2009/

- Snepvangers, J. J. J. C., Heuvelink, G. B. M., & Huisman, J. A. (2003). Soil water content interpolation using spatio-temporal kriging with external drift. Geoderma, 112, 253–271. https://doi.org/10.1016/S0016-7061(02)00310-5

4. Eq. 5: I am not sure why you fit a Beta distribution to the quantiles: isn't the Normal Score Transformation (NST) designed to work with empirical distributions?

   A1.4: We used the Beta-distribution due to its definition on the interval [0,1], thus avoiding conditioning of the resulting distribution (from interpolation) at the extremes.

5. Eq.6 and Eq.7: You applied the NST, so isn't this $E[F(U)] = m$ and same for Variance? Maybe I'm missing something

   A1.5: Yes, you could rewrite Eq.6 and Eq.7 by using $E[U(F_x(Z(\mathbf{x},t))]$ instead. However, to our opinion, it does not make a difference.

6. Nowhere is explained how you calculate the variograms for all the interpolations you do. Maybe worth mentioning it somewhere.

   A1.6: We add the following sentence at the end of the paragraph (p.8, l.4):

   *[…]. The corresponding variograms are calculated using Kendall's tau for a robust interpolation (Lebrenz et al. 2017). […]*

   *Reference:*
   - Lebrenz, H. & Bárdossy, A. (2017). Estimation of the variogram using Kendall's tau for a robust geostatistical interpolation. Journal of Hydrological Engineering, 22(9), 04017038. 10.1061/(ASCE)HE.1943-5584.0001568

7. Pg 7 top: you introduce the elevation dataset, but you don't explain why. Mention you use it for both EDK of the parameters and for the reference EDK of the rainfall process.

   A1.7: We extend the adding the additional sentence at p.7, l.2:

   *[…] The upscaled elevation ultimately serves as external drift for EDK of the parameters within QK and for the reference EDK with the original variable. […]*

8. is the dry ratio the number of stations that recorded zero rainfall over the whole month over the total number of stations? Can you state this a bit more explicitly?

   A1.8: We rephrase the sentence at p.7,l.9 in order to clarify the definition of the "dry ratio":

   *[…] The observed average monthly precipitation over the twelve calendar months c is illustrated in Fig.3 along with the percentage of zero-value observations over all observations of the specific calendar month c (hereafter referred to as dry ratio), revealing a seasonal variation. […]*

9. P.7, l. 18: if you fit a PDF for each month for each station, you have only 22 points to do it, it seems a very little number to be statistically robust. maybe one of the reasons why you need to fit mean and variance rather than the parameters?

   A1.9: The 22 points might be a contributing factor but we rather believe that the resulting (very small) parameters $\vartheta_{2,c}$ are an outcome from the extrapolation.

10. Eq. 8 and eq. 9: you here present both the distributions but don't explain why. Do you want to compare their performance? How did you select Gamma and Weibull distributions? Nowhere in the paper you comment on which one performs best overall.

   A1.10: We used Gamma & Weibull – distributions as exemplary distribution, because:
   1. they are both defined on the interval $[0, \infty]$;
   2. they are frequently used for the variable of monthly distribution;

3. they have only 2 parameters to be interpolated.

The intention of this paper is not to evaluate the distributions but rather to implement the general idea of Quantile Kriging. However, we agree on the inclusion of a statement on which one preforms best (see A1.16)

11. P.8, l.27: You need to state that you do EDK with elevation as the drift. One of the problems I have in this comparison is that often EDK is performed with radar data, which probably would do better than elevation in defining the spatial pattern of rainfall. Can you comment on this?

A1.11: We didn't use radar data for two reasons: First, the availability of radar data is limited in South Africa: they are only available for a relatively short time, limited to urban centers and are not preprocessed/converted into rainfall sums. Secondly, radar images might be useful for real-time predictions but not for long-time (i.e. monthly or yearly) sums, where they show strong systematical errors (Pfaff, 2013).

*Reference:*
- Pfaff, T. (2013). Processing and analysis of weather radar data for use in hydrology. Ph.D. Thesis, Institute for Modelling Water and Environmental Systems, University of Stuttgart, http://dx.doi.org/10.18419/opus-487

12. P.9, l. 19: One of the drawbacks I observe is that QK does not estimate a higher uncertainty where there are less rain gauges, eg. top left corner of Figure 5f.

A1.12: Since the entire area (e.g. top left corner, Fig. 5f) shows the same standard deviation $\sigma_K$, the estimation uncertainty appears to be less dependent from the position of the raingauges.

13. Explain what rho (eq. 12) represent, why you use it, what is its range, and what the optimal value)

A1.13: We use the Spearman rank correlation $\rho_S$ as a non-parametric measure to describe the monotonic relation between estimator $Z^*$ and estimation standard deviation $\sigma_K$, instead of the standard Pearson correlation coefficient, describing only the linear relation. We add the following explanation at p.9, l.26:

*[…]. The non-parametric Spearman rank correlation $\rho_S$ describes the monotonic relation between the estimator $Z^*$ and estimation standard deviation $\sigma_K$, ranging from -1 (negative) to + 1 (positive) with 0 indicating its absence. […].*

14. P 13: I find the explanation about chi squared a bit confusing. I could not understand what had to be uniform and why, until later on you introduce the histogram. Maybe worth introducing the histograms first? or at least explain more in details.

A1.14: Yes, we agree: the explanation could be more precise. We will explain in more detail by rewording the existing explanation on p.13, l.14 by:

*[…]. The test on uniformity verifies the estimated, conditional distribution $F_{Z^*}$ by calculating its value $F_{Z^*}(z(x_i, t))$ for every original observation $z(x_i, t)$. The resulting values (or quantiles) should be uniformly distributed on the interval [0,1] (Bárdossy and Li, 2008). […].*

15. Conclusions: You need to write more here, and remove one of the two paragraphs that are repeated (l. 20-26 or 27-3).

A1.15: Yes, they are actually repeating and we remove the first paragraph (p.14, ll.20-26) and write more in an additional, subsequent paragraph (see A1.16)

16. I feel in general a little bit more discussion of the overall results could be introduced either in the Results and Discussion or the Conclusion section, including many of the comments I mentioned before.

A1.16: We include an additional paragraph at p.15,l.4, including comments from above:

*[…]. The variable of monthly precipitation, observed at 226 raingauges over 264 consecutive time steps, serves as input data. We selected the two parametric $\Gamma$-distribution and Weibull distribution, because they are defined on the interval $[0, \infty]$ and are suitable to describe the variable of monthly precipitation. The selected distributions are fitted to the observations of a specific calendar month, implying an absence of temporal dependence between two sample members (e.g. between the monthly precipitation of December 2002 and December 2003). However, QK does accommodate temporal independence between consecutive observations, unlike existing spatio-temporal Kriging methods. In general, other types of distributions, with a higher number of parameters could be selected, especially in case of other variables of interest. Finally, we used elevation as external drift, both for the interpolation of the parameters within QK as well as for the reference EDK. […].*

And add the following sentences into the last paragraph:

*at p.15,l.6: […] In case of the estimator, QK-$\Gamma$ performs slightly better than QK-Wei for most of the selected objective functions. […].*

*at p.15,l.8: […] In general, QK-Wei shows a superior estimation of the associated uncertainty than QK-$\Gamma$. […].*

---

## Author Comment (AC2) · 14 Aug 2018

**Responses on the Referees 3 comments on the submitted manuscript "Geostatistical interpolation by Quantile Kriging" hess-2018-276**

We are very thankful for your remarks and reply on them below:

**RC 3:**
The article presents an interesting approach to kriging with skewed variables and to non-stationarity: i) For every single location the distribution over time is estimated and quantiles are estimated. ii) To the quantiles of a given time-step a Beta-distribution is fit-ted. iii) The quantiles of the Beta-distribution are transformed by a Normal-Score trans-formation into standard Gaussian variables. iv) Ordinary kriging of the transformed variables. v) Backtransformation of the kriging results to the original scale
One to my opinion main result now is that the variance of the prediction is dependent on the data values themselves, too, and not as in ordinary kriging only dependent on the kriging location. The methodology reminds me somewhat to trans-Gaussian kriging, where you have a similar effect, with the difference that you are still stationary. Maybe you could a little bit comment on this and also on the relationship to copulas.

> A3.1: The interpolation of the beta distributed distribution function values can be seen as a Trans-Gaussian Kriging. Trans-Gaussian Kriging (Spöck et al., 2009) can also be interpreted as a Gaussian copula based linear interpolation.
>
> *Reference:*
> - Spöck, G., Kazianka, H. & Pilz, J. (2009). Modeling and Interpolation of Non-Gaussian Spatial Data: A Comparative Study. Dept. of Statistics, Alpen-Adria Universität, Klagenfurt.
>   https://www.stat.aau.at/Tagungen/statgis/2009/StatGIS2009_Spoeck_2.pdf

Non-stationarity comes into play because you estimate at each spatial location the quantiles separately. You calculate quantiles, and quantiles are always related to copulas,- is there also here a relationship to copulas? Please, elaborate on that. I am also not completely sure, why you need the Beta-distribution at all and not directly calculate the Normal-Score transformation.

> A3.2: compare to reply on comment 4 from Referee 1: we used the Beta-distribution due to its definition on the interval [0,1], thus avoiding conditioning of the resulting distribution at the extremes of 0 and 1.

---

## Author Comment (AC3) · 14 Aug 2018

**Responses on the comments of Referee 2 on the submitted manuscript "Geostatistical interpolation by Quantile Kriging "hess-2018-276**

We thank the reviewer for the valuable comments. Instead of going into detail with the individual points we give a description of the model, which could explain the questionable details.

**Description of the Process**

Obviously, there is an underlying process assumption behind the model. A sketchy description is as follows:

Let $Y_0(x,t)$ be independent (for each different $t$) normal stationary spatial fields with $E[Y_0] = 0$ and $D^2(Y_0) = 1$ for each $t$.

Let:

$$Y_1(x,t) = Y_0(x,t) + M(t) \tag{1}$$

where $M(t)$ is a process (in time) with zero mean. We may assume that the distribution of $M(t)$ is normal. In this case each $x$ $Y_1(x,t)$ is normally distributed with $N(0,d)$. Further for each $t$ the distribution of $Y_1(x,t)$ is $N(M(t),1)$ . Now $Y_2$ is defined as:

$$Y_2(x,t) = \Phi_{0,d}(Y_1(x,t)) \tag{2}$$

where $\Phi_{0,d}$ is the distribution function of $N(0,d)$. By definition $0 \leq Y_2(x,t) \leq 1$

The rainfall is then *generated* as:

$$Z(x,t) = F_x^{-1}(Y_2(x,t)) \tag{3}$$

where $F_x$ is the distribution function of rainfall at location $x$. The $F_x$s can be different due to topography and other influencing factors. These $F_x$-s can be interpolated - example see also in Mosthaf and Bardossy (2017).

We use $Y_2(x,t)$ for each $t$ and assume that it follows a beta distribution. In fact its distribution depends on $M(t)$. If $M(t) = 0$ for all $t$-s then monthly rainfall is fully characterised by independent realizations over space. In this case the distribution of $Y_2$ is uniform for each $t$.

This however is not the case with observed data. The reason is that wet and dry conditions occur simultaneously over the whole area. This is controlled by $M(t)$. One can take $M(t)$ for example as independent random variables or as an ARMA process. If $M(t) \neq 0$ then the distribution of $Y_2(x,t)$ for this $t$ is not uniform. The reason for assuming it as beta was due to the fact that beta distributions are very flexible and can well describe distributions in $[0,1]$. The exact form of the corresponding distribution would be something like:

$$G_t(v) = \Phi_{0,1}\left(\Phi_{M(t),1}^{-1}(v)\right)$$

However the use of this would require the exact knowledge of $M(t)$ for each $t$. We decided to use a simple beta distribution instead.

The introduction of $M(t)$ is reasonable as it explains the difference between the correlation between stations and the spatial correlation calculated using a variogram type approach for a given time. The later correlations are usually lower (smaller ranges) which are increased by the common large scale weather described with $M(t)$.

In our procedure we start with $Z(x,t)$, estimate and interpolate $F_x$. The calculate $Y_2$ for the observation locations. We interpolate $Y_2$ and come back to $Z(x,t)$.

Spatial variograms are calculated for $Y_2$ for each $t$, and $Y_2$ is stationary in space. Non-stationarity and non-Gaussian distributions occur only for $Z$. That is the reason why we concentrate on $Y_2$.

[Figure]

Abbildung 1: Example of the histogram of the observed quantiles in February 1989, along with the fitted Beta-distribution

Reference:

Mosthaf, T. and A. Bárdossy, Regionalizing non-parametric precipitation amount models on different temporal scales, *Hydrology and Earth System Sciences*, **21** , 2463-2481, 2017

---

## Author Comment (AC4) · 2 Oct 2018

**Response to reviewer 2**

We thank the reviewer for the valueable comments. We provide a detailed point by point answer to the reviewers remarks.

The authors propose a new kriging technique developed to regionalize non-normally distributed spatio-temporal variables. The approach, as I understand it, involves (i) fitting time series observations to a predetermined distribution type at each gauge independantly; (ii) using the fitted distribution to assign a probability of non-exceedance to each observation at each gauge; (iii) fit the subset of these probabilities corresponding to each observation time to (different) beta distributions; (iv) map the fitted probabilities to normal quantiles; (v) use the (now normally distributed and \*assumed\* second-order stationary) outcomes as a basis to apply OK or EDK, as described (rather cryptically) in section 2.1.2. While the approach is intriguing, its description in the paper lacks statistical rigor and minimal proofs and intuitions. Its application in cross validation suggests that it does a decent job at predicting some measure of rainfall, but so would most properly fitted interpolation methods. This does not mean that the estimator is BLUE (best  or even efficient  unbiased linear estimator) of the considered stochastic process. Actually, the apparent correlation between Z\*(x) and sigma2(x) (Figure 4) suggests that the process has some degree of heteroskedasticity.  Under these conditions, linear models (like most kriging estimators) are not necessarily efficient (see, for instance, the Gauss Markov Theorem for ordinary least squares). Again, the approach is promising, but its exposition needs a major overhaul to be convincing.

The reviewer described the procedure we applied reasonably well.  However note that the distributions fitted for each location are the same type (gamma for example). The parameters of the distribution have to be interpolated, this step is missing from the reviewers description.

An intuitive description of the procedure is based on the following properties of precipitation fields:

- The monthly (and daily) precipitation amounts for a given month often follow a skewed distribution.

- Monthly (even daily) precipitation amounts cannot be considered as stationary in space. Differences in expected precipitation amounts become clear if one considers long time accumulations.

- The precipitation generating meteorological processes are usually of large spatial extent. This means if there is heavy rainfall at one location it likely that other locations also have heavy rainfall.

- Correlations between time series of precipitation indicate a strong spatial dependence, while the spatial dependence of precipitation on a given time accumulations (day, month) usually show a much weaker spatial dependence.

A possible process model reflecting the above properties can be described as follows:

Let $Y_0(x,t)$ be independent (for each different $t$) normal stationary spatial fields with $E[Y_0] = 0$ and $D^2(Y_0) = 1$ for each $t$.

In order to reflect large scale meteorological processes the process $M(t)$ is introduced. High $M(t)$ values correspond to heavy rainfall covering the area - while low correspond to dry conditions. This $M$ modifies the spatial process:

$$Y_1(x,t) = Y_0(x,t) + M(t) \tag{1}$$

Were $M(t)$ is a process (in time) with zero mean. We may assume that the distribution of $M(t)$ is normal. In this case for each $x$ $Y_1(x,t)$ is normally distributed with $N(0,d)$ with $d = \sqrt{1 + \sigma_M^2}$.

For each fixed $t$ the distribution of $Y_1(x,t)$ is $N(M(t),1)$ . Now $Y_2$ is temporal non-exceedence probability at location $x$ - formally:

$$Y_2(x,t) = \Phi_{0,d}(Y_1(x,t)) \tag{2}$$

where $\Phi_{0,d}$ is the distribution function of $N(0,d)$. (By definition $0 \le Y_2(x,t) \le 1$.)

The rainfall is then *generated* as:

$$Z(x,t) = F_x^{-1}(Y_2(x,t)) \tag{3}$$

where $F_x$ is the distribution function of rainfall at location $x$. The $F_x$s can be different for different $x$ locations due to topography and other influencing factors. (These $F_x$-s can be interpolated - example see also in Mosthaf and Bardossy (2017)).

We use $Y_2(x,t)$ for each $t$ and assume that it follows a beta distribution. In fact its distribution depends on $M(t)$. If $M(t) = 0$ for all $t$-s then monthly rainfall is fully characterised by independent realizations over space. In this case the distribution of $Y_2$ is uniform for each $t$.

This however is not the case with observed data. The reason is that wet and dry conditions occur simultaneously over the whole area. This is controlled by $M(t)$. One can take $M(t)$ for example as independent random variables or to follow an ARMA process. If $M(t) \neq 0$ then the distribution of $Y_2(x,t)$ for this $t$ is not uniform. The reason for assuming it as beta was due to the fact that beta distributions are very flexible and can well describe distributions in $[0,1]$. The exact form of the corresponding distribution would be something like:

$$G_t(v) = \Phi_{0,1}\left(\Phi_{M(t),1}^{-1}(v)\right)$$

However the use of this would require the estimation of $M(t)$ for each $t$. We decided to use a simple beta distribution instead.

The introduction of $M(t)$ is reasonable as it explains the difference between the correlation between stations and the spatial correlation calculated using a variogram type approach for a given time. The later correlations are usually lower (smaller ranges) which are increased by the common large scale weather described with $M(t)$. Note that the introduction of $M(t)$ leads to a correlation of the precipitation time series even if the individual *snapshots* $Y_0(x, t)$ are independent in space.

In our procedure we start with $Z(x, t)$, estimate and interpolate $F_x$. Than calculate $Y_2$ for the observation locations. We interpolate $Y_2$ and come back to $Z(x, t)$.

Spatial variograms are calculated for $Y_2$ for each $t$, and $Y_2$ is stationary in space. Non-stationarity and non-Gaussian distributions occur only for $Z$. That is the reason why we concentrate on $Y_2$.

Non-Gaussianity: In its canonical form, Ordinary Kriging is based on the method of moment (i.e. variance minimization subject to unbiasedness) and so is not technically restricted to normally distributed processes. Gaussiannity is, however, required for maximum likelihood (ML) type estimations, which some studies have shown can be more efficient for kriging-based regionalization (e.g., Lark 2000). ML approaches become necessary to have unbiased prediction of both the mean and the variance when it comes to EDK or Universal Kriging, particularly when subject to more intricate error correlation structures (e.g., Muller 2015). Since non-gaussianity appears to be a key rationale for the proposed approach, it is important to be specific on that.

Non-Gaussianity is considered because of the usually skewed distribution of precipitation amounts for a given time step. The suggested model should enable a simulation of the precipitation amounts. Non-Gaussianity is only in the sense of the marginal distribution at a given time-step. The spatial dependenceis considered to correspond to a multi-Gaussian copula. This kind of transformation is frequently used - for example for Lognormal Kriging.

Beta-distribution: The use of the beta distribution is intriguiging, but more intuition is needed on the assumed underlaying stochastic process. Please describe clearly the properties of the (space-time) stochastic process that you assume gives rise to the observed sample and use that to demonstrate Eqn 6 and 7 in a rigorous mathematical proof. I find the use of the beta distribution promising because a common interpretation is that it describes the distribution of the probabilities associated to a binomial process observed over a finite sample. Lets say that the binomial process in question is the exceedance of a given threshold (as eluded to in the manuscript). Then, if the underlaying point process is identically distributed in space and if an identically sized sample is taken at each observation point, the proportion of observations lower than a given

threshold across all gauges will be beta-distributed. Perhaps thats a start?

The above description shows that the beta distribution is only a convenient tool, not a statistically rigorous approach. As the beta distribution is very flexible it provided an easy and quick approximation of the distribution of the non exceedence probabilities.

Stationarity: More fundamentally, a main issue that I have is that your approach involves fitting distributions independantly at different points in space (gamma or weibull) and time (beta), which implies that the underlaying random point process follows a different distribution at each point in space. Granted, EDK and Universal kriging allow the first moment of the underlaying distribution to vary through space, as allowed by the scaling properties of the expected value estimator. However, I am not aware of any existing geostatistical approach that allows for higher order moments to vary through space. This may be fine, but please demonstrate that your approach does not violate the second order stationarity assumption (i.e that the variogram is constant through space), which is critical (and arguably more important than gaussianity) for kriging.

The distributions fitted to the individual locations are supposed to have a spatial dependence. Further they are assumed to follow the same distribution. These distributions are then interpolated. Here we use the assumption that the distributions show a much clearer effect of the large scale rainfall generating meteorological processes than a sigle mothly (or daily) realization would show. Therefore the use of extrenal covariates , such as topography is more appropriate for this interpolation. The use of these distributions transforms the process to a staionary one. The stationary process is interpolated using the beta distribution of the non-exceedance probabilities. The reason for this is that we intended to avoid problems with interpolated probabilities being outside the $[0, 1]$ interval.

We intend to to revise the paper and to include the above description and discussions.

Mosthaf, T. and A. Bárdossy, Regionalizing non-parametric precipitation amount models on different temporal scales, *Hydrology and Earth System Sciences*,**21** , 2463-2481, 2017

---

## Referee Report (RR1)

Observed

$$Z \xrightarrow[\text{temporal distribution}]{\text{Fit}} \Theta \xrightarrow[\substack{\text{corresponding} \\ \text{to } Z}]{\text{Quantile}} \omega \xrightarrow[\text{transform}]{\beta} \alpha, \beta \xrightarrow[\text{transform}]{\text{Norm}} U$$

Interpolated

$$\hat{Z} \quad \circ \text{——} \quad \hat{\Theta} \quad \hat{\omega} \quad \text{——} \quad \hat{\alpha} \, \hat{\beta} \quad \circ \text{——} \quad \hat{U}$$

PCA OK

OK

---

## Author Response (AR2)

**hess-2018-276– Responses and minor revisons**

We thank the editor and the reviewer for their comments and the suggested, minor changes. Please find below the original comments and our responses.

**Reviewers Comment Nr.1**

*The authors have satisfactorily addressed my concerns from the last submission. I particularly appreciate your effort to explain the intuition behind the approach and the assumed underlying stochastic processes (now in Section 2.2). This makes the whole paper much clearer (and, I believe, more enjoyable to read), and allows the reader to better appreciate the depth of this paper's contribution. This is now a very nice paper and I am looking forward to seeing it published in HESS. I have a few minor suggestions that I think would improve the clarity of the paper:*

*1. The approach proposed is complex with numerous steps and intermediate variables. Perhaps a flow chart would be useful in (i) identifying the letter-variables (e.g., so we immediately know that u refers to the gaussian quantiles when referred to in the text) and (ii) providing a map of what transformation applies to which variable. I have attached a rough sketch of what I have in mind.*

**Response:**

We have added a flowchart corresponding to the suggested flowchart at the end of paragraph 2.1 (see: *"hess-2018-276_QuantileKriging_markedup.pdf"*: p.7., l.5).

**Reviewers Comment Nr.2**

*2. In the introduction, you rightly point out that temporal variations are a key feature of hydrologic variables, as opposed to the much more static geology context in which kriging was originally developed. However, when I think "hydrology", I immediately think streamflow. Of course, I realize that your paper is about rainfall, not streamflow, but I think it would be worth mentioning (here or perhaps in the conclusion) that the approach you propose is limited to spatially distributed variables. This includes rainfall, but not streamflow, which is a spatially aggregated variable that is distributed along a stream network. For this, other geostatistical approaches that account for river network topologies can be referred to (e.g., Skoien et al 2006, Muller and Thompson 2015 and Cressie et al 2006). Perhaps an interesting avenue for further research would be to extend you approach, which accounts for temporal variations, to stream networks (?).*

[revised manuscript text omitted]